# Rewarded soups: towards Pareto-optimal alignment by interpolating weights fine-tuned on diverse rewards

Alexandre Ramé[* 1]  Guillaume Couairon[† 1 2]  Corentin Dancette[† 1]  Jean-Baptiste Gaya[† 1 2]  Mustafa Shukor[† 1]
Laure Soulier[1]  Matthieu Cord[1 3]

## Abstract

Foundation models are first pre-trained on vast unsupervised datasets and then fine-tuned on labeled data. Reinforcement learning, notably from human feedback (RLHF), can further align the network with the intended usage. Yet the imperfections in the proxy reward may hinder the training and lead to suboptimal results; the diversity of objectives in real-world tasks and human opinions exacerbate the issue. This paper proposes embracing the heterogeneity of diverse rewards by following a multi-policy strategy. Rather than focusing on a single a priori reward, we aim for Pareto-optimal generalization across the entire space of preferences. To this end, we propose *rewarded soup*, first specializing multiple networks independently (one for each proxy reward) and then interpolating their weights linearly. This succeeds empirically because we show that the weights remain linearly connected when fine-tuned on diverse rewards from a shared pre-trained initialization. We demonstrate the effectiveness of our approach for text-to-text (summarization, helpful assistant), text-image (image captioning, text-to-image generation, visual grounding), and control (locomotion) tasks. We hope to enhance the alignment of deep models, and how they interact with the world in all its diversity.

## 1. Introduction

Foundation models (Bommasani et al., 2021) have emerged as the standard paradigm to learn neural networks' weights. They are typically first pre-trained through self-supervision (Devlin et al., 2019; Brown et al., 2020; Caron et al., 2021; Radford et al., 2021) and then fine-tuned (Oquab et al., 2014; Yosinski et al., 2014) via supervised learning (Vapnik, 1999). Yet, collecting labels is expensive, and thus supervision may not cover all possibilities and fail to perfectly align (Amodei et al., 2016; Taylor et al., 2016; Ngo, 2022) the trained network with the intended applications. Recent works (Stiennon et al., 2020; Ouyang et al., 2022; Pinto et al., 2023) showed that deep reinforcement learning (DRL) helps by learning from various types of rewards. A prominent example is reinforcement learning from human feedback (RLHF) (Stiennon et al., 2020; Christiano et al., 2017; Ziegler et al., 2019; Wu et al., 2021), which appears as the current go-to strategy to refine large language models (LLMs) into conversational agents such as ChatGPT (Ouyang et al., 2022; OpenAI, 2023). After pre-training on next token prediction (Radford et al., 2018), the LLMs are fine-tuned to follow instructions (Wei et al., 2022; Wang et al., 2022c; Taori et al., 2023) before reward maximization. This RL strategy enhances alignment by evaluating the entire generated sentence instead of each token independently, handling the diversity of correct answers and allowing for negative feedback (Goldberg, 2023). Similar strategies have been useful in computer vision (CV) (Pinto et al., 2023; Rennie et al., 2017), for example to integrate human aesthetics into image generation (Lee et al., 2023; Wu et al., 2023b).

**Diversity of proxy rewards.** RL is usually seen as more challenging than supervised training (Dulac-Arnold et al., 2021), notably because the real reward—ideally reflecting the users' preferences—is often not specified at training time. Proxy rewards are therefore developed to guide the learning, either as hand-engineered metrics (Papineni et al., 2002; Lin & Hovy, 2003; Vedantam et al., 2015) or more recently in RLHF as models trained to reflect human preferences (Christiano et al., 2017; Kwon et al., 2023; Xu et al., 2023). Nonetheless, designing reliable proxy rewards for evaluation is difficult. This *reward misspecification* (Amodei et al., 2016; Pan et al., 2022) between the proxy reward and the users' actual rewards can lead to unforeseen consequences (Michaud et al., 2020). Moreover, the diversity of objectives in real-world applications complicates the challenge. In particular, human opinions can vary significantly (Wildavsky, 1987; Coello, 2000; Schwartz et al., 2012) on subjects such as aesthetics (Nadal & Chat-

---

[*]Project lead, main contributor,[†]Equal experimental contribution [1]Sorbonne Université, CNRS, ISIR, Paris, France [2]Meta AI [3]Valeo.ai. Correspondence to: Alexandre Ramé <alexandre.rame@isir.upmc.fr>.

*ILHF Workshop ICML 2023*.

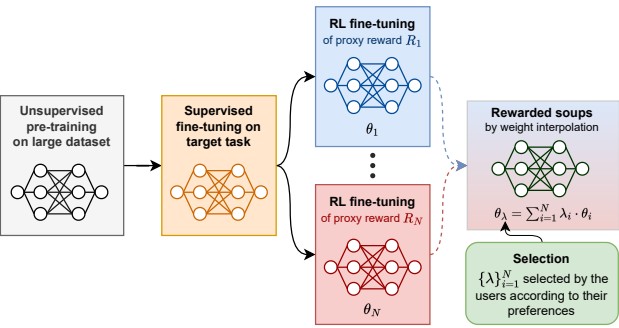

Figure 1: We detail the steps in rewarded soup (RS). After unsupervised pre-training and supervised fine-tuning, we launch $N$ independent RL fine-tunings on the proxy rewards $\{R_i\}_{i=1}^N$. Then we combine the trained networks by interpolation in the weight space. The final weights are adapted at test time by selecting $\lambda$.

terjee, 2019), politics or fairness (Lopez-Paz et al., 2022). Humans have also different expectations from machines: for example, while (Bai et al., 2022a) stressed aligning LLMs towards helpful, honest, and harmless (Askell et al., 2021) feedback, others' interests are to make LLMs mostly engaging and enjoyable (Irvine et al., 2023). Even hand-engineered metrics can be in tension: generating shorter descriptions with higher precision can increase the BLEU (Papineni et al., 2002) score but decrease the ROUGE (Lin & Hovy, 2003) score due to reduced recall.

**Towards multi-policy strategies.** Considering these challenges, it may not be feasible to develop a single model simultaneously aligned with everyone's preferences (Ouyang et al., 2022). Current strategies tend to align towards a consensus-based user (Bakker et al., 2022; Ovadya, 2023), inherently prioritizing certain values over others, potentially resulting in unfair representations of marginalized groups (Kirk et al., 2023). Moreover, these trade-offs (Pan et al., 2023) are decided a priori before training, shifting the responsibility to the engineers and reducing transparency and explainability (Hayes et al., 2022). These limitations, further discussed in Appendix A.2.1, highlight a key limitation of single-policy alignment strategies; their inability to handle the diversity of human preferences. Yet, "human-aligned artificial intelligence is a multi-objective problem" (Vamplew et al., 2018). Thus, we draw inspiration from the multi-objective reinforcement learning (MORL) literature (Barrett & Narayanan, 2008; Li et al., 2020; Tanaka & Yamamura, 2003; Van Moffaert & Nowé, 2014; Roijers et al., 2013; Rădulescu et al., 2020) and notably (Hayes et al., 2022) arguing that tackling diverse rewards requires shifting from single-policy to multi-policy approaches. As optimality depends on the relative preferences across those rewards, the goal is not to learn a single network but rather a **set of Pareto-optimal networks** (Pareto, 1964).

In this paper, we propose **rewarded soup** (RS), an efficient and flexible multi-policy strategy to fine-tune any foundation model. As shown in Section 1, we first use RL to learn one network for each proxy reward; then, we combine these expert networks according to user preferences. This a posteriori selection allows for better-informed trade-offs, improved transparency and increased fairness (Hayes et al., 2022; Mannion et al., 2021). The method to combine those networks is our main contribution: we do this through **linear interpolation in the weight space**, despite the non-linearities in the network. This is in line with recent findings on linear mode connectivity (LMC) (Frankle et al., 2020; Neyshabur et al., 2020): weights fine-tuned from a shared pre-trained initialization remain linearly connected and thus can be interpolated. This LMC inspired a plethora of weight interpolation (WI) strategies (Wortsman et al., 2022a; Ramé et al., 2022; Matena & Raffel, 2022; Ilharco et al., 2022; Don-Yehiya et al., 2022; Ramé et al., 2023), discussed in Section 4. Actually, the name *rewarded soups* follows the terminology of *model soups* (Wortsman et al., 2022a), as we combine various *ingredients* each rewarded differently. Unlike previous works, which focused on supervised learning, we explore LMC in RL, in a challenging setup where each training run uses a different reward. Perhaps surprisingly, we show that we can trade off the capabilities of multiple weights in a single final model, thus without any computational overhead. This enables the creation of custom weights for any preference over the diverse rewards.

- We propose a new practical strategy named rewarded soup for fine-tuning foundation models with diverse rewards. It defines a continuous set of (close to) Pareto-optimal solutions by weight interpolation, approximating more costly multi-policy strategies.

- We analyze the linear mode connectivity between weights fine-tuned on diverse rewards.

- We validate that RS mitigates reward misspecification.

In Section 3, we demonstrate the consistent effectiveness of rewarded soup across a variety of tasks: RLHF fine-tuning of LLaMA, multimodal tasks such as image captioning, text-to-image generation with diffusion models or visual grounding, as well as locomotion tasks. More results are on our anonymized website.

## 2. Rewarded soups

### 2.1. RL fine-tuning with diverse rewards

We consider a deep neural network $f$ of a fixed non-linear architecture (e.g., with batch normalization (Ioffe & Szegedy, 2015), ReLU layers (Agarap, 2018) or self-attention (Vaswani et al., 2017)). It defines a policy by

mapping inputs $x$ to $f(x, \theta)$ when parametrized by $\theta$. For a reward $\hat{R}$ and a test distribution $T$ of deployment, our goal is to maximize $\int_{x \in T} \hat{R}(f(x, \theta))$. Learning $\theta$ is now commonly a three-step process: unsupervised pre-training, supervised fine-tuning, and reward optimization. Yet $\hat{R}$ is usually not specified, meaning we can only optimize a proxy reward $R$ during training. This **reward misspecification** between $R$ and $\hat{R}$ may hinder the alignment of the network with $\hat{R}$. Moreover, the **diversity of human preferences** complicates the design of $R$.

Rather than optimizing one single proxy reward, our paper's first key idea is to consider a family of $N$ diverse proxy rewards $\{R_i\}_{i=1}^N$. Each of these rewards evaluates the prediction according to different (potentially conflicting) criteria. The goal then becomes obtaining a coverage set of policies that trade-off between these rewards. To this end, we first introduce the costly MORL baseline. Its inefficiency motivates our rewarded soups, which leverages our second key idea: weight interpolation.

**MORL baseline.** The standard MORL scalarization strategy (Barrett & Narayanan, 2008; Li et al., 2020) linearizes the problem by interpolating the proxy rewards using $M$ different weightings. Specifically, during the *training phase*, $M$ trainings are launched, with the $j$-th optimizing the reward $\sum_{i=1}^N \mu_i^j R_i$, where $\forall j \in \{1, ..., M\}, \{\mu_i^j\}_{i=1}^N \in \Delta_N$ the $N$-simplex s.t. $\sum_{i=1}^N \mu_i^j = 1$ and $0 \le \mu_i^j \le 1$. Then, during the *selection phase*, the user's reward $\hat{R}$ becomes known and the $j$-th policy that maximizes $\hat{R}$ on some validation dataset is selected. We typically expect to select $j$ such that $\sum_{i=1}^N \mu_i^j R_i \approx \hat{R}$ linearly approximates the user's reward. Finally, this $j$-th weight is used during the *inference phase* on test samples. Yet, a critical issue is that "minor [preference] variations may result in significant changes in the solution" (Vamplew et al., 2008). Thus, a high level of granularity in the mesh of $\Delta_N$ is necessary. This requires explicitly maintaining a large set of $M \gg N$ networks, practically one for each possible preference. Ultimately, this MORL strategy is unscalable in deep learning due to the **computational, memory, and engineering costs** involved (see further discussion in Appendix A.2.2).

**Rewarded soup (RS).** In this paper, we draw inspiration from the weight interpolation literature. The idea is to learn expert weights and interpolate them linearly to combine their abilities. Specifically, we propose RS, illustrated in Section 1 and whose recipe is described below. RS alleviates MORL's scaling issue as it requires only $M = N$ trainings while being flexible and transparent.

1. During the *training phase*, we optimize a set of $N$ expert weights $\{\theta_i\}_{i=1}^N$, each corresponding to one of the $N$ proxy rewards $\{R_i\}_{i=1}^N$, and all from a shared pre-trained initialization.

2. For the *selection phase*, we linearly interpolate those weights to define a continuous set of rewarded soups policies: $\{\sum_{i=1}^N \lambda_i \cdot \theta_i\}_{\{\lambda_i\}_{i=1}^N \in \Delta_N}$. Practically, we uniformly sample $M$ interpolating coefficients $\{\{\lambda_i^j\}_{i=1}^N\}_{j=1}^M$ from the $N$-simplex $\Delta_N$ and select the $j$-th that maximizes the user's reward $\hat{R}$ on validation samples, i.e., $\text{argmax}_{j=1}^M \hat{R}\left(\sum_{i=1}^N \lambda_i^j \theta_i\right)$.

3. For the *inference phase*, we predict using the network $f$ parameterized by $\sum_{i=1}^N \lambda_i^j \theta_i$.

**While MORL interpolates the rewards, RS interpolates the weights.** This is a considerable advantage as the appropriate weighting $\lambda$, which depends on the desired trade-off, can be selected *a posteriori*; the selection is achieved without additional training, only via inference on some samples. In the next Section 2.2 we explicitly state the Hypotheses 1 and 2 underlying in RS. These are considered *Working Hypotheses* as they enabled the development of our RS strategy. Their empirical verification will be the main motivation for our experiments on various tasks in Section 3.

## 2.2. Exploring the properties of the rewarded soups set of solutions

### 2.2.1. LINEAR MODE CONNECTIVITY OF WEIGHTS FINE-TUNED ON DIVERSE REWARDS

We consider $\{\theta_i\}_{i=1}^N$ fine-tuned on $\{R_i\}_{i=1}^N$ from a shared pre-trained initialization. Previous works (Frankle et al., 2020; Neyshabur et al., 2020; Wortsman et al., 2022a; Ramé et al., 2023) defined linear mode connectivity (LMC) w.r.t. a single performance measure (e.g., accuracy or loss) in supervised learning. We extend this notion in RL with $N$ rewards, and define that the LMC holds if all rewards for the interpolated weights exceed the interpolated rewards. It follows that the LMC condition which underpins RS's viability is the Hypothesis 1 below.

**Working Hypothesis 1** (LMC). *For all $\{\lambda_i\}_i \in \Delta_N$ and $k \in \{1, ..., N\}$, $R_k(\sum_i \lambda_i \cdot \theta_i) \ge \sum_i \lambda_i R_k(\theta_i)$.*

### 2.2.2. PARETO OPTIMALITY OF REWARDED SOUPS

The Pareto front (PF) is the set of undominated weights, for which no other weights can improve a reward without sacrificing another, i.e., $\{\theta \mid \nexists \theta' \in \Theta \text{ s.t. } \{R_i(\theta')\}_{i=1}^N >_N \{R_i(\theta)\}_{i=1}^N\}$ where $>_N$ is the dominance relation in $\mathcal{R}^N$. In practice, we only need to retain one policy for each possible value vector, i.e., a Pareto coverage set (PCS). We now introduce the key Hypothesis 2.

**Working Hypothesis 2** (Pareto optimality). *The set $\{\sum_i \lambda_i \cdot \theta_i | \{\lambda_i\}_i \in \Delta_N\}$ is a PCS of $\{R_i\}_i$.*

Hypothesis 2 holds if the rewarded soups solutions, uncovered by interpolation, are Pareto-optimal. Overall, we

empirically validate Hypotheses 1 and 2 in Section 3, yet also report a few limitations in Appendix and research directions to fix them. Moreover, we theoretically prove in Appendix B.2 they approximately hold when rewards are replaced by their second-order Taylor expansion with co-diagonalizable Hessians, a simplified setup justifiable when weights remain close.

**Remark 1.** *Hypotheses 1 and 2 rely on a good pre-trained initialization, making RS particularly well-suited to fine-tune foundation models. This is because pre-training prevents the weights from diverging during training (Neyshabur et al., 2020). When the weights remain close, we can theoretically justify Hypotheses 1 and 2 (see Appendix B.2) and, more broadly, demonstrate that WI approximates ensembling (Hansen & Salamon, 1990; Lakshminarayanan et al., 2017) (see Lemma 4). In contrast, the LMC does not hold when training from scratch (Neyshabur et al., 2020).*

**Remark 2.** *Pareto-optimality in Hypothesis 2 is defined w.r.t. a set of weights $\Theta$. Yet, in full generality, improvements in initialization, RL algorithms, data, or specific hyperparameters could enhance performances. In other words, for real-world applications, the true PF is unknown and needs to be defined w.r.t. a training procedure. In this case, $\Theta$ represents the set of weights attainable by fine-tuning within a shared procedure. As such, in Section 3 we analyze Hypothesis 2 by comparing the fronts obtained by RS and scalarized MORL while keeping everything else constant.*

### 2.2.3. CONSEQUENCES OF PARETO OPTIMALITY IF THE USER'S REWARD IS LINEAR IN THE PROXY REWARDS

**Lemma 1** (Reduced reward misspecification in the linear case). *If Hypothesis 2 holds, and for linear reward $\hat{R} = \sum_i \hat{\mu}_i R_i$ with $\{\hat{\mu}_i\}_i \in \Delta_N$, then $\exists \{\lambda_i\}_i \in \Delta_N$ such that $\sum_i \lambda_i \cdot \theta_i$ is optimal for $\hat{R}$.*

The proof outlined in Appendix B.1 directly follows the definition of Pareto optimality. In simpler terms, Lemma 1 implies that if Hypothesis 2 is true, then RS can mitigate reward misspecification. For any preference $\hat{\mu}$, there exists a $\lambda$ such that the $\lambda$-interpolation over weights maximizes the $\hat{\mu}$-interpolation over rewards. In practice, as we will see in Figure 4(a), we can set $\lambda = \hat{\mu}$, or cross-validate $\lambda$ on other samples. Yet, this theoretically holds only for $\hat{R}$ linear over the proxy rewards. This follows the *linear utility functions* setup from the MORL literature (Rădulescu et al., 2020), whose limitations (Vamplew et al., 2008) are discussed in Appendix A.1. This motivates having sufficiently rich and diverse proxy rewards to capture the essential aspects of all possible users' rewards. Despite the lack of theoretical guarantees, we will show in Figures 4(b) and 9 that weight interpolation improves results even for non-linear $\hat{R}$.

## 3. Experiments

In this section we implement RS across a variety of standard learning tasks: text-to-text generation, image captioning, image generation, visual grounding, and locomotion. We use either model or statistical rewards. We follow a systematic procedure. First, we independently optimize diverse rewards on training samples. For all tasks, we employ the default architecture, hyperparameters and RL algorithm; the only variation being the reward used across runs. Second, we evaluate the rewards on the test samples: the results are visually represented in series of plots. Third, we verify Hypothesis 1 by examining whether RS's rewards exceed the interpolated rewards. Lastly, as the true Pareto front is unknown in real-world applications, we present empirical support for Hypothesis 2 by comparing the front defined by RS (sliding $\lambda$ between 0 and 1) to the MORL's solutions optimizing the $\mu$-weighted rewards for $0 \leq \mu \leq 1$ (sometimes only $\mu = 0.5$ for computational reasons).

### 3.1. Text-to-text: LLaMA with diverse RLHFs

Given the significance of RLHF to train LLMs, we begin our experiments with text-to-text generation tasks. Our pre-trained network is LLaMA-7b (Touvron et al., 2023), instruction fine-tuned (Wei et al., 2022; Wang et al., 2022b) on Alpaca (Taori et al., 2023). For RL training with PPO (Schulman et al., 2017), we employ the trl package (von Werra et al., 2020) and the setup from (Beeching et al., 2023) with low-rank adapters (LoRA) (Hu et al., 2022a) for efficiency. We consider the following tasks: summarization (Stiennon et al., 2020; Wu et al., 2021) on two datasets (Reuter news (Ahmed, 2017) in Figure 2(a) and Reddit posts (Völske et al., 2017) in Figure 2(b)), and helpfulness as a conversational assistant (Bai et al., 2022a) in Figures 2(c) and 2(d). To evaluate the generation in the absence of supervision, we utilized $N = 2$ different reward models (RMs) for each task, except in Figure 2(d) where $N = 4$. These RMs were trained on human preferences datasets (Christiano et al., 2017) and all open-sourced on HuggingFace (Wolf et al., 2020). For example in summarization, $R_1$ follows the "Summarize from Human Feedback" paper (Stiennon et al., 2020), while $R_2$ leverages "contrast candidate generation" (Chen et al., 2021). For the assistant task, we rely on diverse RMs from OpenAssistant (Köpf et al., 2023); they differ by their training procedures.

The results are reported in Figure 2. The green front, defined by RS between the two weights specialized on $R_1$ and $R_2$, is above the straight line connecting those two points, validating Hypothesis 1. Second, the front passes through the point obtained by MORL fine-tuning on the average of the two rewards, supporting Hypothesis 2. Moreover, when comparing both full fronts, they have qualitatively the same shape; quantitatively in hypervolume (Yen & He,

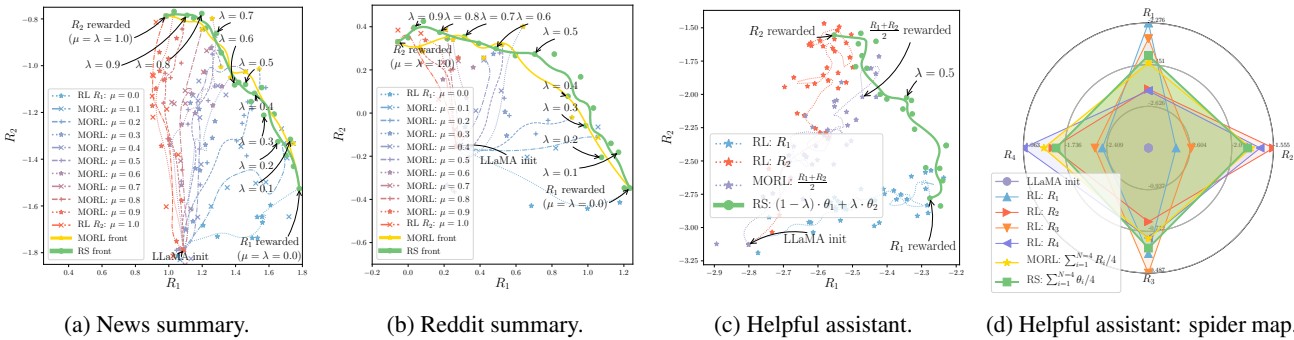

(a) News summary.  (b) Reddit summary.  (c) Helpful assistant.  (d) Helpful assistant: spider map.

Figure 2: RLHF results in NLP with LLaMA-7b (Touvron et al., 2023) and reward models $R_i$ from HuggingFace (Wolf et al., 2020). The blue line reports checkpoints' results along the training trajectory of $\theta_1$ rewarding $R_1$, the red line $\theta_2$ rewarding $R_2$, and the purple line the MORL rewarding $\frac{R_1+R_2}{2}$. Our rewarded soup (RS) linearly interpolates between the weights $\theta_1$ and $\theta_2$; sliding the interpolation coefficient $\lambda$ from 0 to 1 reveals the green solid front of rewarded soups solutions. In Figures 2(a) and 2(b), we additionally show the multiple MORL runs rewarding $(1-\mu) \times R_1 + \mu \times R_2$ with preferences $0 \le \mu \le 1$. It reveals a similar yellow front, yet more costly. In Figure 2(d), we uniformly ($\lambda_i = \frac{1}{4}$) average the weights fine-tuned for the assistant task on $N = 4$ reward models.

2013) (lower is better, the area over the curve w.r.t. an optimal point), RS's hypervolume is 0.367 vs. 0.340 for MORL in Figure 2(a), while it is 1.176 vs. 1.186 in Figure 2(b). Finally, in Figure 2(d), we use $N = 4$ RMs for the assistant task and uniformly average the $N = 4$ weights, confirming that RS can scale and trade-off between more rewards.

### 3.2. Image-to-text: captioning with statistical rewards

RL training is also effective for multimodal tasks (Pinto et al., 2023), for example in image captioning (Rennie et al., 2017) where the task is to generate textual descriptions of images. Precisely evaluating the quality of a prediction w.r.t. a set of human-written captions is a challenging task, thus the literature relies on various hand-engineered, non-differentiable metrics: e.g., the precision-focused BLEU (Papineni et al., 2002), the recall-focused ROUGE (Lin & Hovy, 2003), METEOR (Banerjee & Lavie, 2005) handling synonyms and CIDEr (Vedantam et al., 2015) using TF-IDF. As these metrics are proxies for human preferences, good trade-offs are desirable. We conduct our experiments on COCO (Lin et al., 2014), with an ExpansionNetv2 (Hu et al., 2022b) network and a Swin Transformer (Liu et al., 2022) visual encoder, initialized from the state-of-the-art weights of (Hu et al., 2022b) optimized on CIDEr. We then utilize the code of (Hu et al., 2022b) and their self-critical (Rennie et al., 2017) procedure (a variant of REINFORCE (Williams, 1992)) to reward the network on BLEU1, BLEU4, ROUGE or METEOR. More details are in Appendix D.

We observe in Figure 3 that tuning solely BLEU1 sacrifices some points on ROUGE or BLEU4. Yet interpolating between $\theta_1$ and $\theta_2$ uncovers a convex set of solutions approximating the ones obtained through scalarization of the rewards in MORL. When comparing both full fronts in Figure 3(a), they qualitatively have the same shape, and

quantitatively the same hypervolume (Yen & He, 2013) of 0.140. One of the strengths of RS is its ability to scale to any number of rewards. In Figure 3(c), we uniformly ($\lambda_i = \frac{1}{5}$) average $N = 5$ weights fine-tuned independently. It improves upon the initialization (Hu et al., 2022b) and current state-of-the-art on all metrics, except for CIDEr, on which (Hu et al., 2022b) was explicitly optimized.

Figure 4 refines our analysis of RS. In Figures 4(a) and 4(b), rewards are normalized to 1 for the initialization and 0 for the worst model. Figure 4(a) validates Lemma 1: for any linear preference $\hat{\mu}$ over the proxy rewards, there exists an optimal solution in the set described by RS. Two empirical strategies to set the value of $\lambda$ are close to optimal: selecting $\lambda = \hat{\mu}$ if $\hat{\mu}$ is known, or cross-validating (CV) $\lambda$ if a different data split (Karpathy & Fei-Fei, 2015) is available. Moreover, Figure 4(b) (and Figure 9 in Appendix D) investigate all metrics as evaluation. Excluding results' variance, we observe monotonicity in both training rewards, linear in BLEU1 and quadratic in ROUGE. For other evaluation rewards that **cannot be linearly expressed** over the training rewards, the curves' concavity shows that RS consistently improves the endpoints, thereby mitigating reward misspecification. The optimal $\lambda$ depends on the similarity between the evaluation and training rewards: e.g., best BLEU2 are with small $\lambda$. Lastly, as per (Izmailov et al., 2018) and Lemma 4, Figure 4(c) suggests that RS succeeds because WI approximates *deep ensembling* (Hansen & Salamon, 1990; Lakshminarayanan et al., 2017), interpolating the predictions rather than the weights. Actually, ensembling performs better, but it cannot be fairly compared as its inference cost is doubled.

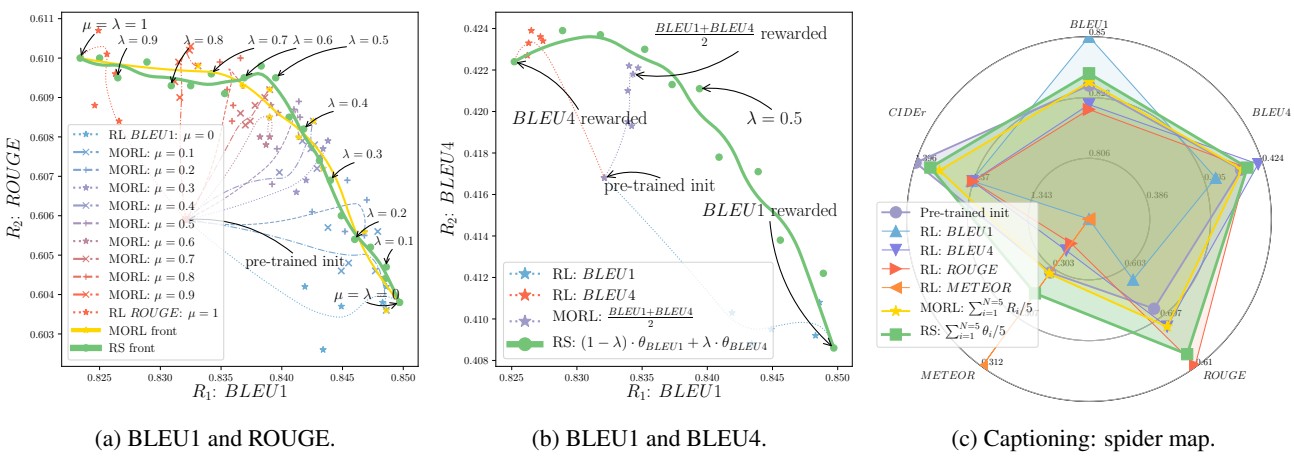

(a) BLEU1 and ROUGE.  (b) BLEU1 and BLEU4.  (c) Captioning: spider map.

Figure 3: Results in image captioning on COCO (Lin et al., 2014). As rewards $R_1$ (blue stars every epoch) and $R_2$ (red stars), we consider standard statistical metrics: BLEU1 (1-gram overlap), BLEU4 (4-grams overlap), ROUGE, METEOR and CIDEr. Figure 3(a) include the MORL training trajectories optimizing $(1 - \mu) \times BLEU1 + \mu \times ROUGE$, uncovering a yellow front similar to RS's green front. In Figure 3(c), RS uniformly averages the 5 weights (one for each reward), resulting in the largest area and the best trade-off.

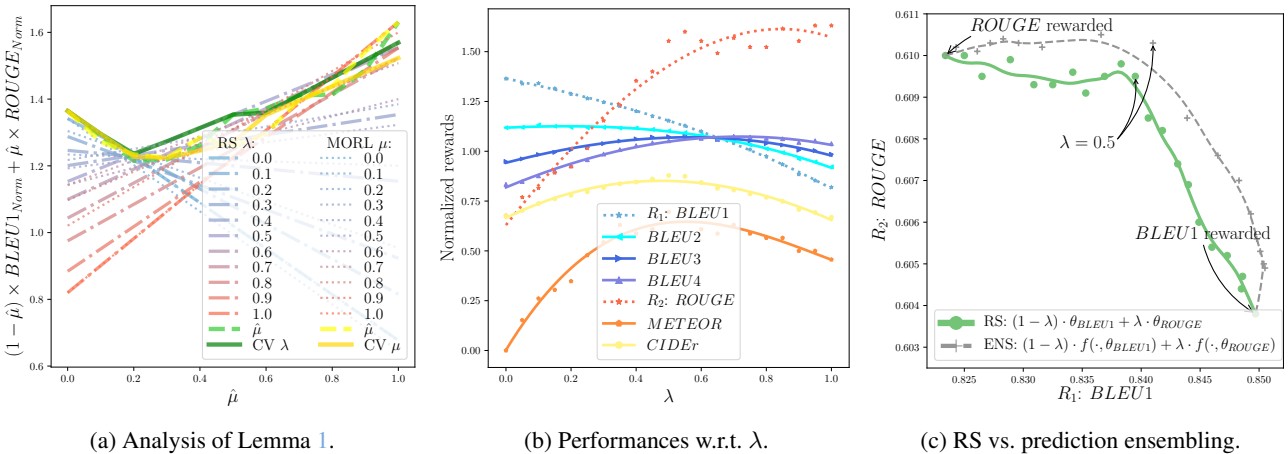

(a) Analysis of Lemma 1.  (b) Performances w.r.t. $\lambda$.  (c) RS vs. prediction ensembling.

Figure 4: Refined results in captioning with $R_1 = BLEU1$ and $R_2 = ROUGE$. Figure 4(a) empirically validates Lemma 1 by reporting results of RS (for varying $\lambda$) and of MORL (for varying $\mu$) for varying user's preference $\hat{\mu}$. In Figure 4(b), all rewards are used for evaluation as a function of the interpolating coefficient. In Figure 4(c), we report the front of the costly ensembling (Hansen & Salamon, 1990; Lakshminarayanan et al., 2017) of predictions (rather than of weights).

### 3.3. Text-to-image: diffusion models with RLHFs

Beyond text generation, we now apply RS to align text-to-image generation with human feedbacks, as previously done in recent papers(Lee et al., 2023; Wu et al., 2023b; Xu et al., 2023). This alignment is expected to improve specific visual control signals like colors, counts, and backgrounds. Notably, diffusion models can be fine-tuned to match human aesthetic preferences. Our network is a diffusion model (Ho et al., 2020) with 2.2B parameters, pre-trained on an internal dataset of 300M images; it reaches similar quality as Stable Diffusion (Rombach et al., 2022), which was not used for copyright reasons. As for any subjective metric, there is a variety of reward models that can capture different aspects

of aesthetic preference. In our experiments, we employ $N = 2$ open-source reward models: *ava*, trained on the AVA dataset (Murray et al., 2012), and *cafe*, trained on a mix of real-life and manga images. These two models are trained in a supervised setting to match human quality ratings collected. We first generate 10000 images; then, for each reward, we remove half of the images with the lowest reward's score, and fine-tune 10% of the parameters (Xie et al., 2023) on the reward-weighted negative log-likelihood (Lee et al., 2023). More details are in Appendix E.

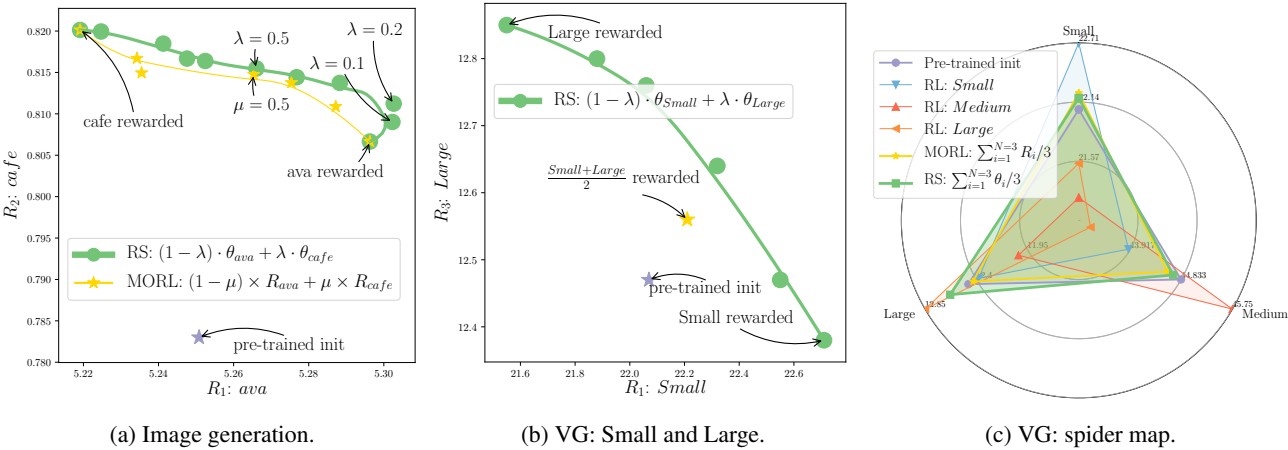

(a) Image generation.          (b) VG: Small and Large.          (c) VG: spider map.

Figure 5: Figure 5(a) reports our RLHF experiments on text-to-image generation with diffusion models. From the pre-trained initialization, we learn $\theta_{ava}$ and $\theta_{cafe}$ by optimizing the two reward models *ava* and *cafe*. Interpolation between them reveals the green Pareto-optimal front, above the yellow MORL front. Figures 5(b) and 5(c) report our results in visual grounding (VG) on RefCOCO+ (Yu et al., 2016), where we optimize to predict boxes with IoU> 0.5 w.r.t. the ground-truth, for objects of either small, medium or large size.

The results displayed in Figure 5(a) validate Hypothesis 1, as the front described by RS when sliding $\lambda$ from 0 and 1 is convex. Moreover, RS gives a better front than MORL, validating Hypothesis 2. Interestingly, the *ava* reward model seems to be more general-purpose than *cafe*, as RL training on *ava* also enhances the scores of *cafe*. In contrast, the model $\theta_{cafe}$ performs poorly in terms of *ava* in Figure 5(a). Nonetheless, RS with $(1 - \lambda) \cdot \theta_{ava} + \lambda \cdot \theta_{cafe}$ outperforms $\theta_{ava}$ alone, not only in terms of *cafe*, but also of *ava* when $\lambda \in \{0.1, 0.2\}$. These findings confirm that RS can better align text-to-image models with a variety of aesthetic preferences. This ability to adapt at test time paves the way for a new form of user interaction with text-to-image models, beyond prompt engineering.

### 3.4. Text-to-box: visual grounding

We now consider visual grounding (VG) (Yu et al., 2016): the task is to predict the bounding box of the region described by an input text. We use a seq-to-seq unified model predicting the box auto-regressively as a sequence of location tokens (Wang et al., 2022a). This model is pre-trained on a large image-text dataset, then fine-tuned with cross-entropy for VG; finally, we use a weighted loss between the cross-entropy and REINFORCE in the RL stage. As the main evaluation metric for VG is the accuracy (i.e., intersection over union (IoU) > 0.5), we consider 3 non-differentiable rewards: the accuracy on small, medium, and large objects. We design this experimental setup because improving results on all sizes simultaneously is challenging, as shown in Figure 5(c), where MORL performs similarly to the initialization. The results in Figure 5(b) confirm that optimizing for small objects degrades performance on large ones; fortunately, interpolating can trade-off. In conclusion,

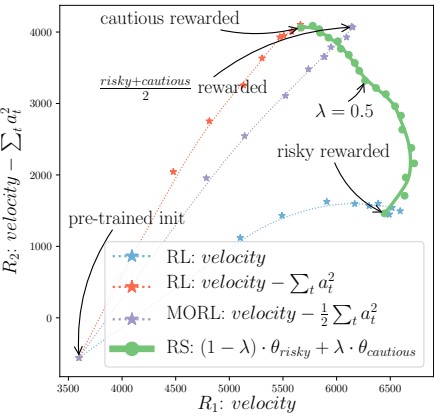

Figure 6: Locomotion results.

we can adapt to users' preferences at test time by adjusting $\lambda$, which in turn changes the object sizes that the model effectively handles. On the one hand, if focusing on distant and small objects, a large coefficient should be assigned to $\theta_{Small}$. On the other hand, to perform well across all sizes, we can recover initialization's performances by averaging uniformly (in Figure 5(c)). More details are in Appendix F.

### 3.5. Locomotion with diverse engineered rewards

Teaching humanoids to walk in a human-like manner (Duan et al., 2016) serves as a benchmark to evaluate RL strategies (Ng et al., 1999) for continuous control. One of the main challenges is to shape a suitable proxy reward (Dorigo & Colombetti, 1994; Dewey, 2014), given the intricate coordination and balance involved in human locomotion. It is standard (Todorov et al., 2012) to consider dense rewards of the

form $R = velocity - \alpha \times \sum_t a_t^2$, controlling the agent's velocity while regularizing the actions $\{a_t\}_t$ taken over time. Yet, the penalty coefficient $\alpha$ is challenging to set. To address this, we devised two rewards in the Brax physics engine (Freeman et al., 2021): a risky $R_1$ with $\alpha = 0$, and a more cautious $R_2$ with $\alpha = 1$.

Like in all previous tasks, RS's front in Figure 6 exceeds the interpolated rewards, as per Hypothesis 1. Moreover, RS's front indicates an effective balance between risk-taking and cautiousness, providing empirical support for Hypothesis 2, although MORL with $\mu = 0.5$ (i.e., $\alpha = 0.5$) slightly surpasses RS's front. For a more qualitative and intuitive assessment, we provide animations of our RL agent's locomotion on our anonymized website. More details are available in Appendix G.

## 4. Related work

Our RS approach leans on two key components from traditional DRL. The first is **proxy rewards**, whose design is challenging. Statistical metrics, the standard in captioning (Rennie et al., 2017) or language translation (Ranzato et al., 2016), are not practical to measure human concepts (Kwon et al., 2023) such as helpfulness (Bai et al., 2022a; Askell et al., 2021). Reward models can be trained via inverse DRL (Ng et al., 2000; Abbeel & Ng, 2004) when supervision from experts is available, otherwise from prediction comparison in recent RLHF works(Stiennon et al., 2020; Ouyang et al., 2022; Christiano et al., 2017). The latest (Kwon et al., 2023; Bai et al., 2022b; Madaan et al., 2023; Scheurer et al., 2023; Sun et al., 2023) further reduce the labeling costs by using the in-context abilities of LLMs.

Second, RS relies on existing **RL algorithms** to maximize the given rewards. RS succeeds with variants of two of the most common, REINFORCE (Williams, 1992) and PPO (Schulman et al., 2017), suggesting it could be applied to others (Go et al., 2023; Yuan et al., 2023). Among the ensembling-like RL strategies (Wang et al., 2010; Mordatch et al., 2015; Rajeswaran et al., 2017) handling multiple policies, some (Parker-Holder et al., 2020; Osa et al., 2022) aim to explicitly increase the diversity, yet never with foundation models nor weight interpolation. Moreover, pre-training could address stability and exploration issues (Xie et al., 2022; Yang et al., 2023; Sekar et al., 2020).

When dealing with multiple objectives in deep learning, the common approach is to combine them into a single reward (Roijers et al., 2013; Rădulescu et al., 2020): (Glaese et al., 2022) multiply the predictions of a preference RM (evaluating factfullness) and a rule RM (detecting rules breaking). The **multi-policy** alternatives (Barrett & Narayanan, 2008; Li et al., 2020; Tanaka & Yamamura, 2003; Van Moffaert & Nowé, 2014) are usually more costly. To reduce the

cost, (Won et al., 2020; Yang et al., 2020) build experts and then train a new network to combine them; (Mossalam et al., 2016; Wilson et al., 2007; Nguyen et al., 2020) share weights across experts; (Castelletti et al., 2013; Yang et al., 2019; Abels et al., 2019; Peschl et al., 2021) directly train a single model; the recent and more similar (Hua et al., 2023) learns one linear embedding per (locomotion) task that can be interpolated. Yet, these works are mostly for academic benchmarks (Todorov et al., 2012; Vamplew et al., 2011); adapting them to larger tasks (e.g., RLHF for foundation models with PPO) is challenging as they modify the training procedure. Finally, we relate to **multitask learning** (Caruana, 1997), where predictions are evaluated for multiple tasks; in contrast, we have a single prediction evaluated by multiple rewards.

Recent works extended the **linear mode connectivity** when fine-tuning on different tasks (Ilharco et al., 2022; Don-Yehiya et al., 2022; Ramé et al., 2023; Wu et al., 2023a) or with different losses (Ramé et al., 2022; Croce et al., 2023), while (Juneja et al., 2023) highlighted some failures in NLP for classification. In contrast, we investigate the LMC in RL. The most similar works are for control system tasks: (Lawson & Qureshi, 2023) averaging decision transformers and (Gaya et al., 2022) explicitly enforcing connectivity in subspaces of policies trained from scratch on a single reward. When the LMC holds, combining networks in weights combines their abilities (Ilharco et al., 2023; Daheim et al., 2023); e.g., averaging an English summarizer and an English-to-French translator can summarize in French (Jang et al., 2023). In domain generalization, (Wortsman et al., 2022a; Ramé et al., 2022; Arpit et al., 2021) showed that WI reduces model misspecification (D'Amour et al., 2020); by analogy, we show that RS reduces reward misspecification.

## 5. Conclusion

As AI systems are increasingly applied to crucial real-world tasks, there is a pressing issue to align them to our specific and diverse needs, while making the process more transparent and limiting the cultural hegemony of a few individuals. In this paper, we proposed rewarded soup, a strategy that efficiently yields Pareto-optimal solutions through weight interpolation after training. Our experiments have consistently validated our working hypotheses for various significant large-scale learning tasks, demonstrating that rewarded soup can mitigate reward misspecification. We hope to inspire further research in exploring how the generalization literature in deep learning can help for alignment, to create AIs that benefit society as a whole.

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

# Rewarded soups: towards Pareto-optimality by interpolating weights fine-tuned on diverse rewards

## Supplementary material

This supplementary material is organized as follows:

- Appendix A further discusses the societal impacts of our RS strategy.

- Appendix B details some theoretical aspects of our RS strategy.

- Appendix C details our experiments in RLHF with LLaMA for text-to-text generation.

- Appendix D details and enriches our experiments in image captioning.

- Appendix E details and enriches our experiments in image generation.

- Appendix F details and enriches our experiments in visual grounding.

- Appendix G details and enriches our locomotion experiments.

The shareable code will be released on this anonymized url page. Moreover, you can find additional qualitative results of our experiments on our anonymized website.

## A. Discussion

### A.1. Limitations and societal impacts

The recent and rapid scaling of networks presents both opportunities and major concerns (Amodei et al., 2016; Hendrycks & Mazeika, 2022; Hendrycks, 2023). Our approach is a step towards better **empirical alignment** (Taylor et al., 2016; Ngo, 2022). Yet, reward misspecification is only one of the many challenges inherited from the RL paradigm. First, proxy rewards may lack robustness (Gao et al., 2022) or be hacked (Skalse et al., 2022) via adversarial exploitation, making them unreliable. Second, RL algorithms may cause overfitting, leading to poor generalization in test, with a risk of goal misgeneralization (Shah et al., 2022; Di Langosco et al., 2022). Third, RLHF has drawbacks, such as harming calibration (OpenAI, 2023). Our a posteriori multi-policy strategy could alleviate the impact of some badly shaped proxy rewards and some failed optimizations, as well as tackling Goodhart's law (Smith, 2021). Yet, without constraint on the test distribution, complete alignment may be impossible (Wolf et al., 2023), for example for LLMs with prompts of arbitrary (long) length. Therefore, new training paradigms (Keskar et al., 2019; Korbak et al., 2023) beyond RL may be required.

**Theoretical guarantees** for alignment are also needed (Rodriguez-Soto et al., 2021). Yet, RS relies on an empirical finding: the LMC (Frankle et al., 2020), which currently lacks full theoretical guarantees, even in the simplest case of moving averages (Izmailov et al., 2018). The best existing explanation (Ramé et al., 2022; Izmailov et al., 2018) relies on the similarities between weight interpolation and functional ensembling (Hansen & Salamon, 1990; Lakshminarayanan et al., 2017) when weights remain close, as recalled in Lemma 4. Moreover, assuming the LMC, Lemma 1 theoretically fixes issues only for $\hat{R}$ linear over the proxy rewards. Yet, such **linearization** cannot encapsulate all types of (human) preferences (Vamplew et al., 2018; 2008). Thus, considering more complex combinations (Gábor et al., 1998; An et al., 2021; Van Moffaert et al., 2013; Smith et al., 2021) is a promising direction. We may empirically overcome this limitation within RS by continually adjusting and adding new proxy rewards, such that their linear mixtures have increasingly good coverage. Indeed, RS is flexible and was shown to handle variable numbers of rewards.

Finally, our a posteriori alignment with users facilitates **personalization** (Salemi et al., 2023) of models. As discussed in Appendix A.2.1 and in (Kirk et al., 2023), this could increase usefulness by providing tailored generation, notably to under-represented groups. Moreover, the distributed nature of RS makes it parallelizable thus practical in a federated

learning setup (McMahan et al., 2017) where data must remain private. Yet, this personalization comes with risks for individuals of "reinforcing their biases […] and narrowing their information diet"(Kirk et al., 2023). This may worsen the polarization of the public sphere. Under these concerns, we concur with the notion of "personalization within bounds" (Kirk et al., 2023), with these boundaries potentially set by weights fine-tuned on diverse and carefully inspected rewards.

## A.2. Benefits of our approach

In this section we discuss the benefits of our rewarded soup (RS) approach with respect to the two families of strategies: the **single-policy** and the **multi-policy** approaches.

### A.2.1. COMPARED TO SINGLE-POLICY APPROACHES

The main reason why single-policy approaches are not suitable is because they optimize over a single set of preferences. In contrast, we build a coverage set of Pareto-optimal policies. This is important for the following reasons, mostly first discussed in Hayes *et al.* (Hayes et al., 2022) and in Kirk *et al.* (Kirk et al., 2023).

Indeed, the user's true reward is highly uncertain before training. This "semi-blind" (Hayes et al., 2022) manual process forces a priori and uncertain decisions about the required trade-offs. It **shifts the responsibility** from the problem stakeholders to the system engineers, who need to anticipate the impact of their choices on the final performance. Critically, the RLHF process may cause the "tyranny of the crowdworker" (Kirk et al., 2023), as models are "tailored to meet the expectations of [...] a small number of crowdworkers primarily based in the US, with little to no representation of broader human cultures, geographies or languages." (Kirk et al., 2023). Moreover, these "biases are exacerbated by a lack of […] documentation" (Kirk et al., 2023). Thus (Kirk et al., 2023) argue that the **personalization should be explicit**—rather than implicitly caused by hidden and chaotic engineering choices. In contrast, our strategy could **support decision-making** to find a good balance between (potentially conflicting) parties' interests. This value pluralism (Tetlock, 1986) can lead to **fairer** and more equitable outcomes (Vamplew et al., 2018; Siddique et al., 2020). Single-policy cannot adapt to test time requirements; in contrast, RS facilitates personalized assistances (Salemi et al., 2023), with fewer prompts/inputs to the model, as we only need to adapt interpolating coefficients and not the full network. This is all the more important as human preferences change from time to time: in this **dynamic utility function** scenario, RS can quickly adapt by adjusting the $\lambda$ to match new preferences. Finally, RS could also improve the **interpretability** and **explainability** of the decisions. Letting the users decide could make the process more **transparent** (Gabriel & Ghazavi, 2021), which is essential to ensure that the development process is fair, unbiased, and inclusive (Abadi et al., 2016).

### A.2.2. COMPARED TO MULTI-POLICY APPROACHES

The main reason why other multi-policy approaches through multitasking are not suitable is because of their **computational costs** required to learn a dense set of policies. In contrast, RS only trains the proxy rewards independently, and enables the selection of the interpolating coefficient a posteriori. This is especially useful with large number of rewards and thus growing number of combinations. Second, multitask (Caruana, 1997) is challenging; for example, even if the true reward is actually a linear weighted sum of some proxy rewards and those coefficients are known, using those preferences during training can lead to suboptimal results (Van Moffaert et al., 2014), because of conflicting gradients (Yu et al., 2020; Liu et al., 2021) or different variance scales (Espeholt et al., 2018; Teh et al., 2017). This has been tackled in RL, but so far mostly for games such as ATARI (Bellemare et al., 2013). Third, our strategy is compatible with the inherent **iterative engineering process** of alignment. Indeed, RS can continually include adjusted opinions while preventing forgetting of the old behaviours. This relates to the **continual learning** challenge, and the empirical observations that weight averaging can reduce catastrophic forgetting (Stojanovski et al., 2022; Eeckt et al., 2022). Moreover, as shown in (Ilharco et al., 2023) and confirmed in Figure 10(c), negative editing by weight interpolation can fix and force the removal of some behaviours. Finally, RS is computationally effective, requiring **no communication across servers**, thus enabling "embarrassingly simple parallelization" (Li et al., 2022). This facilitates its use in **federated learning** scenario (McMahan et al., 2017) where the data should remain private. Actually, RS follows the **updatable machine learning paradigm** (Raffel, 2021), "allowing for the collaborative creation of increasingly sophisticated AI system" (Ramé et al., 2023). In the future, we may develop open-source personalized models, rewarded on decentralized private datasets, and combine them continuously.

# B. Theoretical insights

## B.1. Proof of Lemma 1

*Proof.* Considering $\theta$ maximizing $\hat{R}$, we first show that $\theta$ is on the PF of $\{R_i\}_i$. Otherwise, considering $\theta' >_N \theta$ and as $\forall i, \hat{\mu}_i \geq 0$, we have $\sum_i \hat{\mu}_i R_i(\theta') > \sum_i \hat{\mu}_i R_i(\theta)$. This implies that $\theta'$ would produce a better policy than $\theta$ for $\hat{R} = \sum_i \hat{\mu}_i R_i$ and thus the contradiction. Finally, as $\theta$ is on the PF and by definition of a PCS, there exists $\lambda$ s.t. $\forall k, R_k(\sum_i \lambda_i \cdot \theta_i) = R_k(\theta)$. $\qquad\square$

## B.2. Theoretical guarantees with quadratic rewards

In this section, we provide theoretical guarantees for the near-optimality of RS when considering quadratic rewards. This simplification amounts to replacing the rewards by their second-order Taylor approximation, which is a realistic assumption when the weights remain within a small neighborhood.

### B.2.1. SIMPLE CASE WITH HESSIANS PROPORTIONAL TO THE IDENTITY MATRIX

For the first Lemma 2, we make the following simplifying Assumption 1.

**Assumption 1** (Hessians proportional to the Identity matrix.)**.** *Every reward $R_i$ is quadratic, with Hessians proportional to $\mathbb{I}_d$. Specifically, let $\Theta \subset \mathbb{R}^d$ be the set of possible weights, and let $\{R_i\}_{i=1}^N$ be the $N$ rewards, we can write for $i \in \{1, ..., N\}$:*

$$\forall \theta \in \Theta, \quad R_i(\theta) = R_i(\theta_i) - \eta_i \|\theta - \theta_i\|^2 \tag{1}$$

*where $\eta_i \in \mathbb{R}_+^*$ and $\theta_i$ is the global maximum for reward $R_i$.*

**Lemma 2.** *Let $\hat{\mu} = (\hat{\mu}_1, ..., \hat{\mu}_N) \in \Delta_N$. Then, under Assumption 1, the reward $R_{\hat{\mu}} = \sum_i \hat{\mu}_i \times R_i$ is maximized on the convex hull of $\{\theta_1, ..., \theta_N\}$.*

*Proof.* The function $R_{\hat{\mu}}$ is quadratic thus has an unique global maximum $\hat{\theta}$, that we find analytically:

$$\nabla_\theta R_{\hat{\mu}}(\hat{\theta}) = 0 \implies \sum_{i=1}^N \mu_i \eta_i \cdot (\hat{\theta} - \theta_i) = 0$$

$$\implies \hat{\theta} = \frac{\sum_{i=1}^N \hat{\mu}_i \eta_i \cdot \theta_i}{\sum_{i=1}^N \hat{\mu}_i \eta_i}$$

Since all the $\hat{\mu}_i \eta_i$ are positive or zero, and at least one is greater than zero, $\hat{\theta}$ is indeed in the convex hull of $\{\theta_1, ..., \theta_N\}$. $\qquad\square$

**Remark 3.** *Under Assumption 1, the reward functions are concave; thus we can reasonably assume that each fine-tuning procedure for $R_i$ reaches its global optimum $\theta_i$ for $i \in \{1, ..., N\}$. Then, Lemma 2 tells us that the maximum value for linear user's reward $R_{\hat{\mu}}$ is obtainable by weight interpolation between the $\{\theta_i\}_{i=1}^N$: the interpolating coefficients in $\Delta_N$ such that $\lambda_i \propto \hat{\mu}_i \eta_i$ make rewarded soups optimal.*

### B.2.2. ADVANCED CASE WITH DIAGONAL HESSIANS

We now consider the more complex case with the relaxed Assumption 2. For simplicity, we only consider $N = 2$ rewards $R_1$ and $R_2$.

**Assumption 2** (Diagonal Hessians)**.** *The rewards are quadratic, with Hessians diagonal negative definite. Specifically, we can write for $i \in \{1, 2\}$:*

$$\forall \theta = (\theta^1, ..., \theta^d) \in \Theta, \quad R_i(\theta) = R_i(\theta_i) - \sum_{j=1}^d \eta_i^j (\theta^j - \theta_i^j)^2, \tag{2}$$

*where $(\eta_i^1, ... \eta_i^d) \in \{\mathbb{R}_+^*\}^d$ and $\theta_i = (\theta_i^1, ..., \theta_i^d)$ is the global maximum for reward $R_i$.*

**Remark 4.** *This diagonal Assumption 2 of the Hessian is common: for example in optimization (LeCun et al., 2012; Kingma & Ba, 2015), to prune networks (LeCun et al., 1990) or in out-of-distribution generalization (Rame et al., 2022). This strong assumption is supported by the empirical observation (Becker & Le Cun, 1988) that Hessians are diagonally dominant, in particular at the end of training. Also, we note that our findings remain valid assuming only that the Hessians are co-diagonalizable.*

**Lemma 3.** *We consider the user's reward $R_{\hat{\mu}} = (1 - \hat{\mu}) \times R_1 + \hat{\mu} \times R_2$ with $\hat{\mu} \in [0, 1]$, and*

$$\Delta R_{\hat{\mu}} = \max_{\theta \in \Theta} R_{\hat{\mu}}(\theta) - \max_{\lambda \in [0,1]} R_{\hat{\mu}}((1 - \lambda) \cdot \theta_1 + \lambda \cdot \theta_2). \tag{3}$$

*$\Delta R_{\hat{\mu}}$ corresponds to the difference in terms of $R_{\hat{\mu}}$ between the global maximum and the maximum reachable by weight interpolation through rewarded soups (with a single interpolating coefficient for all dimensions). Then, under Assumption 2, we have:*

$$\Delta R_{\hat{\mu}} \leq \frac{\hat{\mu}^2 (1 - \hat{\mu})^2 (M\Delta_1 - \Delta_2)(M\Delta_2 - \Delta_1)}{(\hat{\mu}(1 - \hat{\mu})(M - 1)^2 + M)((1 - \hat{\mu})\Delta_1 + \hat{\mu}\Delta_2)}, \tag{4}$$

*where $M = \max_{j \in \{1,...,d\}} \max\left(\frac{\eta_1^j}{\eta_2^j}, \frac{\eta_2^j}{\eta_1^j}\right)$ is the maximum of eigenvalues ratio, $\Delta_1 = R_1(\theta_1) - R_1(\theta_2)$ and $\Delta_2 = R_2(\theta_2) - R_2(\theta_1)$.*

*When $\Delta_1 = \Delta_2$, the bound simplifies into:*

$$\Delta R_{\hat{\mu}} \leq \frac{\hat{\mu}^2 (1 - \hat{\mu})^2 (M - 1)^2}{\hat{\mu}(1 - \hat{\mu})(M - 1)^2 + M} \Delta_1 \tag{5}$$

*Furthermore, when the Hessians are equal, then $M = 1$ and $\Delta R_{\hat{\mu}} = 0$: RS is optimal .*

*Proof.* This novel proof is in three steps. First, we find $\hat{\theta}$ maximizing $R_{\hat{\mu}}(\theta)$ for $\theta$ on the full set of weights $\Theta$. Second, we find $\bar{\lambda}$ maximizing $R_{\hat{\mu}}((1 - \lambda) \cdot \theta_1 + \lambda \cdot \theta_2)$ for $\lambda \in [0, 1]$ and thus defining the best interpolation between the expert weights. Finally, we bound $\Delta R_{\hat{\mu}}$, the differences between their rewards, by applying the Bhatia-Davis inequality.

**First step.** Let's first find the maximum of $R_{\hat{\mu}}$ on $\Theta$. Denoting $S = (1 - \hat{\mu}) \times R_1(\theta_1) + \hat{\mu} \times R_2(\theta_2)$, we have for all $\theta \in \Theta$:

$$R_{\hat{\mu}}(\theta) = S - \sum_{j=1}^{d} \left( (1 - \hat{\mu})\eta_1^j \left(\theta^j - \theta_1^j\right)^2 + \hat{\mu}\eta_2^j \left(\theta^j - \theta_2^j\right)^2 \right) \tag{6}$$

Since $R_{\hat{\mu}}$ is a sum of concave quadratic functions, it has a unique global maximum reached at a point we note $\hat{\theta} = \left(\hat{\theta}^1, ..., \hat{\theta}^d\right)$. The global maximum can be computed by differentiating $R_{\hat{\mu}}$ with respect to each variable $\theta^j$, which gives:

$$\hat{\theta}^j = \left(1 - \hat{\lambda}^j\right) \cdot \theta_1^j + \hat{\lambda}^j \cdot \theta_2^j$$

where the interpolating coefficients per dimension $\hat{\lambda}^j$ are defined for $j \in \{1, ..., d\}$ as:

$$\hat{\lambda}^j = \frac{\hat{\mu}\eta_2^j}{(1 - \hat{\mu})\eta_1^j + \hat{\mu}\eta_2^j} \in [0, 1]. \tag{7}$$

**Second step.** With $\lambda \in [0, 1]$ and $\theta = (1 - \lambda) \cdot \theta_1 + \lambda \cdot \theta_2$, we can write $R_{\hat{\mu}}(\theta)$ as a function of $\lambda$:

$$R_{\hat{\mu}}(\theta) = S - \sum_{j=1}^{d} \left( \left((1 - \hat{\mu})\eta_1^j + \hat{\mu}\eta_2^j\right)\left(\lambda - \hat{\lambda}^j\right)^2 + \frac{\hat{\mu}(1 - \hat{\mu})\eta_1^j \eta_2^j}{(1 - \hat{\mu})\eta_1^j + \hat{\mu}\eta_2^j} \right)\left(\theta_1^j - \theta_2^j\right)^2$$

$$= R_{\hat{\mu}}(\hat{\theta}) - \sum_{j=1}^{d} p_j \left(\lambda - \hat{\lambda}^j\right)^2 \tag{8}$$

where $p_j$ is defined as $p_j = \left( (1 - \hat{\mu}) \eta_1^j + \hat{\mu} \eta_2^j \right) \left( \theta_1^j - \theta_2^j \right)^2$.

From Equation (8), we can compute the maximum reward obtainable for weight averaging $\max_{\lambda \in [0,1]} R_{\hat{\mu}}((1 - \lambda) \cdot \theta_1 + \lambda \cdot \theta_2)$. Since the function $\lambda \mapsto R_{\hat{\mu}}((1 - \lambda) \cdot \theta_1 + \lambda \cdot \theta_2)$ is a concave quadratic function, there is a unique value $\bar{\lambda}$ maximizing $R_{\hat{\mu}}$ equal to

$$\bar{\lambda} = \frac{\sum_{j=1}^d p_j \hat{\lambda}^j}{\sum_{j=1}^d p_j}. \tag{9}$$

Since all $p_j$ are positive and all $\hat{\lambda}^j$ are between 0 and 1, $\bar{\lambda}$ is also between 0 and 1. Therefore, $R_{\hat{\mu}}\left( (1 - \bar{\lambda}) \cdot \theta_1 + \bar{\lambda} \cdot \theta_2 \right)$ is indeed the maximum reward for rewarded soups.

**Third step.** Applying Equation (8) to $\bar{\lambda}$ gives:

$$\Delta R_{\hat{\mu}} = R_{\hat{\mu}}(\hat{\theta}) - R_{\hat{\mu}}\left( (1 - \bar{\lambda}) \cdot \theta_1 + \bar{\lambda} \cdot \theta_2 \right) \tag{10}$$

$$= \sum_{j=1}^d p_j \left( \bar{\lambda} - \hat{\lambda}^j \right)^2 \tag{11}$$

$$= \left( \sum_{j=1}^d \frac{p_j}{\sum_{i=1}^n p_i} \left( \bar{\lambda} - \hat{\lambda}^j \right)^2 \right) \left( \sum_{j=1}^n p_j \right) \tag{12}$$

The second term in Equation (12) can be simplified as:

$$\sum_{j=1}^d p_j = (1 - \hat{\mu}) \Delta_1 + \hat{\mu} \Delta_2. \tag{13}$$

The core component of this proof is the upper bounding of the first term in Equation (12). The key idea is to recognize the variance of a discrete random variable $\Lambda$ with $\mathbb{P}(\Lambda = \hat{\lambda}_i) = \frac{p_i}{\sum_{j=1}^n p_j}$; then, $\bar{\lambda}$ from Equation (9) is actually the expectation of $\Lambda$. Then, we can apply the **Bhatia-Davis inequality**, as recalled in Equation (14), on the variance of a bounded random variable $a \leq \Lambda \leq b$:

$$Var(\Lambda) \leq (b - \mathbb{E}(\Lambda))(\mathbb{E}(\Lambda) - a) \tag{14}$$

Therefore Equation (12) is bounded by:

$$\Delta R_{\hat{\mu}} \leq \left( \max_{1 \leq j \leq d} \hat{\lambda}^j - \bar{\lambda} \right) \left( \bar{\lambda} - \min_{1 \leq j \leq d} \hat{\lambda}^j \right) ((1 - \hat{\mu}) \Delta_1 + \hat{\mu} \Delta_2). \tag{15}$$

Now, we bound the variables $\hat{\lambda}^j$, since $1/M \leq \eta_1^j / \eta_2^j \leq M$. Then for all $j$ we have:

$$\frac{\hat{\mu}}{(1 - \hat{\mu}) M + \hat{\mu}} \leq \hat{\lambda}^j \leq \frac{\hat{\mu} M}{(1 - \hat{\mu}) + \hat{\mu} M}, \tag{16}$$

and thus:

$$\Delta R_{\hat{\mu}} \leq \left( \frac{\hat{\mu} M}{1 + \hat{\mu}(M - 1)} - \bar{\lambda} \right) \left( \bar{\lambda} - \frac{\hat{\mu}}{M - \hat{\mu}(M - 1)} \right) ((1 - \hat{\mu}) \Delta_1 + \hat{\mu} \Delta_2). \tag{17}$$

Finally, noting that $\Delta_i = \sum_{j=1}^d \eta_i^j \left( \theta_2^j - \theta_1^j \right)^2$, we deduce from Equation (9) that $\bar{\lambda} = \frac{\hat{\mu} \Delta_2}{(1 - \hat{\mu}) \Delta_1 + \hat{\mu} \Delta_2}$. Replacing this in the previous Equation (17) gives the final Equation (4), concluding the proof. $\qquad \square$

**Remark 5.** *As a final remark, please note that the suboptimality of RS comes from the need of having one single interpolating coefficient $\bar{\lambda}$ for all $d$ parameters $(\theta^1, \ldots, \theta^d)$ of the network. Yet, the advanced merging operations in (Matena & Raffel, 2022) remove this constraint, with interpolating coefficients proportional to the eigenvalues of the Fisher matrices (Fisher, 1922), which actually approximate the eigenvalues of the Hessian (Schraudolph, 2002; Thomas et al., 2020). Combining (Matena & Raffel, 2022) and our RS is a promising research direction, the key issue being the computation of the Fisher matrices (Kunstner et al., 2019) for networks with billions of parameters.*

### B.2.3. BOUND VISUALIZATION

We visualize in Figure 7 the bound given by Lemma 3. We show that for small values of $M$ like $M = 2$, the value of $R_{\hat{\mu}}$ for RS is quite close to the global optimum. Also, recall that RS theoretically matches this upper bound when $M = 1$. For larger values like $M = 10$, the bound is less tight, and we note that the maximum value of $R_{\hat{\mu}}$ approaches the constant function 1 as $M \to \infty$.

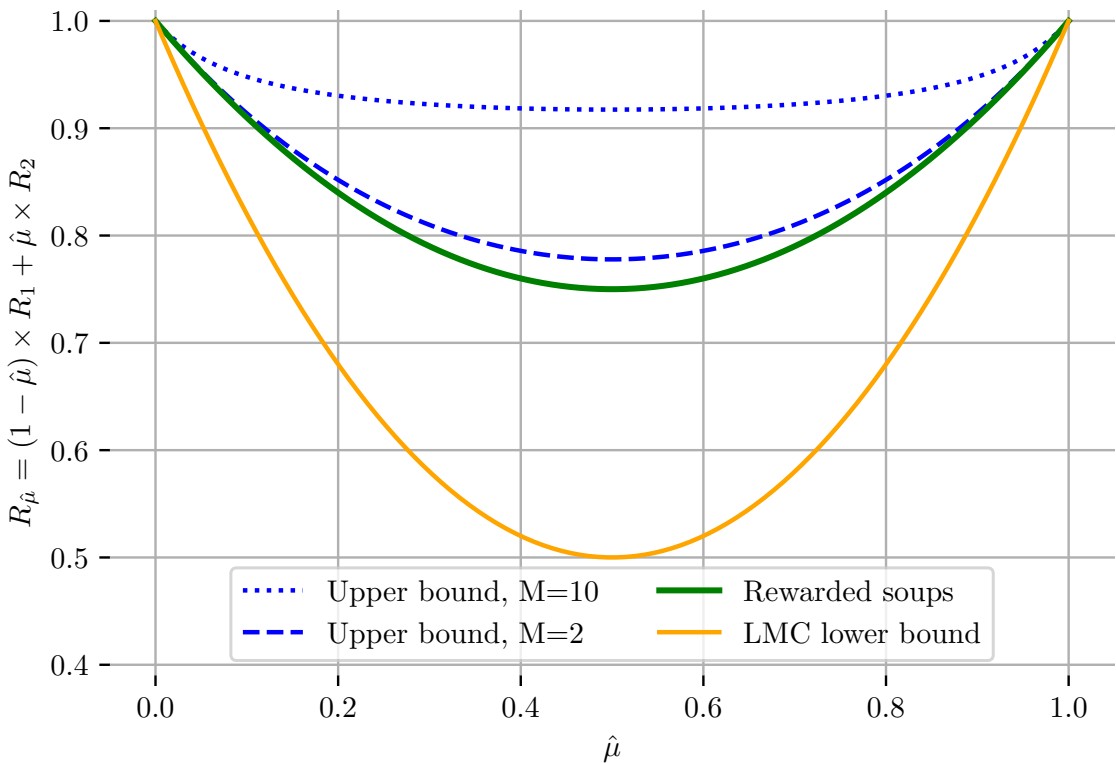

Figure 7: Illustration of the bound given by Lemma 3 under Assumption 2. For simplicity, we showcase the case where $R_1(\theta_1) = R_2(\theta_2) = 1$, $R_1(\theta_2) = R_2(\theta_1) = 0$, thus $\Delta_1 = \Delta_2 = 1$. In green, we plot the rewards obtained with rewarded soups for the optimal $\bar{\lambda}$, i.e., $R_{\hat{\mu}}\left((1 - \bar{\lambda}) \cdot \theta_1 + \bar{\lambda} \cdot \theta_2\right)$, whose value is independent of $M$ in this case. In blues, we plot the maximum value of $\mathcal{R}_{\hat{\mu}}$ given by Equation (5) in Lemma 3, for $M = 2$ and $M = 10$. For reference, we also plot the values for the lower bound in the LMC Hypothesis 1, i.e., equal to $(1 - \hat{\mu})(1 - \bar{\lambda})R_1(\theta_1) + \hat{\mu}\bar{\lambda}R_2(\theta_2)$. As RS outperforms this lower bound, it validates Hypothesis 1 in this case.

### B.3. Similarity between weight interpolation and functional ensembling

**Lemma 4** ($\lambda$-interpolation of weights approximates the $\lambda$-ensembling of predictions. Adapted from (Wortsman et al., 2022a; Ramé et al., 2022; Izmailov et al., 2018).)**.** *Given $\theta_1$ and $\theta_2$ optimized for $R_1$ and $R_2$ s.t. they remain close, i.e., $\|\theta_1 - \theta_2\|_2 \approx 0$. Denoting $\theta_\lambda$ the interpolated weights $\theta_\lambda = (1 - \lambda) \cdot \theta_1 + \lambda \cdot \theta_2$ and $f_\lambda$ the ensembling of predictions $f_\lambda(\cdot) = (1 - \lambda) \cdot f(\cdot, \theta_1) + \lambda \cdot f(\cdot, \theta_2)$:*

$$f(\cdot, \theta_\lambda) \approx f_\lambda(\cdot)$$

*and for $k \in \{1, 2\}$:*

$$R_k(f(\cdot, \theta_\lambda)) \approx R_k(f_\lambda(\cdot))$$

*Proof.* This proof follows (Ramé et al., 2022) and has two components.

**Functional approximation.** First, we perform a Taylor expansion at the first order of the models' predictions w.r.t. parameters $\theta$ for $x \in T$:

$$f(x, \theta_1) = f(x, \theta_\lambda) + \nabla_\theta f(x, \theta_\lambda)^\intercal (\theta_1 - \theta_\lambda) + \mathcal{O}\Big(\|\theta_1 - \theta_\lambda\|_2^2\Big)$$
$$= f(x, \theta_\lambda) + \nabla_\theta f(x, \theta_\lambda)^\intercal (\lambda \cdot \theta_1 - \lambda \cdot \theta_2) + \mathcal{O}\Big(\|\theta_1 - \theta_2\|_2^2\Big)$$

and similarly:

$$f(x, \theta_2) = f(x, \theta_\lambda) + \nabla_\theta f(x, \theta_\lambda)^\intercal ((\lambda - 1) \cdot \theta_1 + (1 - \lambda) \cdot \theta_2) + \mathcal{O}\Big(\|\theta_1 - \theta_2\|_2^2\Big)$$

Then by $\lambda$-weighted sum over $i$, the term multiplying $\nabla_\theta f(x, \theta_\lambda)^\intercal$ cancels out and we obtain:

$$f_\lambda(x) = (1 - \lambda) \cdot f(x, \theta_1) + \lambda \cdot f(x, \theta_2) = f(x, \theta_\lambda) + \mathcal{O}\Big(\|\theta_1 - \theta_2\|_2^2\Big). \tag{18}$$

**Reward approximation.** Second, we obtain the reward approximation with a Taylor expansion at the zeroth order of the reward $R_k$ for $k \in \{1, 2\}$ and injecting Equation (18):

$$R_k(f_\lambda(x)) = R_k(f(x, \theta_\lambda)(x)) + \mathcal{O}(\|f_\lambda(x) - f(x, \theta_\lambda)\|_2)$$
$$= R_k(f(x, \theta_\lambda)(x)) + \mathcal{O}\Big(\|\theta_1 - \theta_2\|_2^2\Big).$$

We obtain the results when $\theta_1$ and $\theta_2$ remain close, i.e., when we can ignore the $\mathcal{O}$ term. $\qquad\square$

## C. Text-to-text: LLaMA with diverse RLHFs

We summarize the key implementation details of our text-to-text generation experiments in Table 1. The pre-trained network is LLaMA-7b (Touvron et al., 2023); then low-rank adapters (Hu et al., 2022a) were fine-tuned on Alpaca (Taori et al., 2023) to follow instructions. We eventually fine-tune via PPO on the different considered tasks. Our code is adapted from (Beeching et al., 2023); we kept most of their hyperparameter values, only dividing by 2 the batch size to fit in our GPU and extending the output length. For each considered task, we downloaded the reward models from HuggingFace (Wolf et al., 2020). For example in summarization tasks, $R_1$ was open-sourced in an effort to reproduce the Summarize from Human Feedback paper (Stiennon et al., 2020), while $R_2$ (Chen et al., 2021) aimed at improved "faithfulness in abstractive summarization with contrast candidate generation". For the assistant task, we rely on different reward models from OpenAssistant (Köpf et al., 2023). Though they all aim at evaluating whether an answer is adequate given a question, they differ in their predictions due to differences in their architecture and training procedures. In practice, we simply leverage them as block-box classification pipelines, implemented in the transformers library (Wolf et al., 2020).

Table 1: LLaMA with RLHF experiments: key implementation details.

| | |
|---|---|
| **Model** | |
| Architecture | Transformer (Vaswani et al., 2017) |
| Pre-training | LLaMA-7b (Touvron et al., 2023) |
| Instruction FT | Alpaca (Taori et al., 2023) |
| **RL procedure** | |
| Fine-tuning strategy | LoRA (Hu et al., 2022a) |
| | *following Alpaca-LoRA (Wang, 2023)* |
| LoRA alpha | 16 |
| LoRA dropout | 0.05 |
| | *following trl-peft (von Werra et al., 2020; Beeching et al., 2023)* |
| Optimizer | Adam (Kingma & Ba, 2015) |
| Learning rate | 1.41e-5 |
| Batch size | 128 |
| Output length | Between 16 and 32 |
| RL algorithm | PPO (Schulman et al., 2017) |
| KL PPO | 0.05 for summary tasks else 0.2 |
| Epochs | 2 for Reuter summary else 1 |
| Hardware | NVIDIA RTX A6000 49 Go |
| Compute budget | 4000 GPUh |
| Task name | **Reuter summary** |
| Description | Generate a concise and clear summary of newspaper articles from Reuters. |
| Prompt | "Generate a one-sentence summary of this post." |
| Dataset | Reuter news from (Ahmed, 2017; Ahmed et al., 2018) from news-summary |
| $R_1$ | gpt2-reward-summarization trained here. |
| $R_2$ | bart-faithful-summary-detector (Chen et al., 2021) |
| Figure | Figure 2(a) |
| Task name | **Reddit summary** |
| Description | Generate a concise and clear summary of posts from Reddit across a variety of topics (subreddits). |
| Prompt | "Generate a one-sentence summary of this post." |
| Dataset | Reddit crawl from the TL;DR dataset (Völske et al., 2017) from summarize-from-feedback (Stiennon et al., 2020) |
| $R_1$ | gpt2-reward-summarization trained here. |
| $R_2$ | bart-faithful-summary-detector (Chen et al., 2021) |
| Figure | Figure 2(b) |
| Task name | **Helpful assistant** |
| Description | Provide helpful and harmless answers to potentially complex and sensitive questions. |
| Prompt | No prompt, only users' questions. |
| Dataset | Helpfulness and Harmlessness datasets (Bai et al., 2022a) from hh-rlhf |
| $R_1$ | reward-model-deberta-v3-large-v2 |
| $R_2$ | reward-model-electra-large-discriminator |
| $R_3$ | reward-model-deberta-v3-base-v2 |
| $R_4$ | reward-model-deberta-v3-base |
| Figure | Figures 2(c) and 2(d) |

# D. Image-to-text: captioning with diverse rewards

## D.1. Experimental details

We summarize the key implementation details of our captioning experiments in Table 2. In short, we took the state-of-the-art network (Hu et al., 2022b) for captioning on COCO, fine-tune with their code and only changing the reward. In more details, since the *self-critical* paper (Rennie et al., 2017) (a variant of REINFORCE (Williams, 1992) with a specific estimation of the baseline score) it is now common in captioning to optimize the CIDEr reward (Vedantam et al., 2015) after a first step of supervised fine-training. The recent ExpansionNetv2 (Hu et al., 2022b) follows this strategy to reach state-of-the-art results, with a Swin Transformer (Liu et al., 2022) visual encoder and a block static expansion for efficiency. We investigate whether additional RL trainings can help. We use their code and most of their hyperparameters, only reducing the batch size from 24 to 18 to fit in our GPUs and consequently adapt the learning rate.

Table 2: Captioning experiments: key implementation details.

| | |
|---|---|
| **Model** | |
| Architecture | ExpansionNetv2 (Hu et al., 2022b) |
| Visual encoder | Swin Transformer (Liu et al., 2022) |
| Visual encoder pre-training | ImageNet 22k (Deng et al., 2009) |
| Fine-tuning | Cross-entropy then CIDEr RL (Rennie et al., 2017) on COCO (Lin et al., 2014) |
| **RL procedure** | |
| Fine-tuning strategy | Usually frozen visual backbone, but end-to-end in Figure 10(d) |
| RL algorithm | self-critical (Rennie et al., 2017), a variant of REINFORCE (Williams, 1992) |
| Optimizer | Radam (Liu et al., 2020) |
| Dataset | COCO (Lin et al., 2014) and Karpathy split (Karpathy & Fei-Fei, 2015) |
| Rewards | BLEU (with 1-gram or 4-grams), ROUGE, METEOR, CIDEr |
| Learning rate | 1e-5 |
| Batch size | 18 |
| Gradient accumulation | 2 |
| Warmup | Anneal 0.8 during 1 epoch |
| Epochs | 6 |
| Hardware | GPU V100 32G |
| Compute budget | 1500 GPUh |

## D.2. Additional results

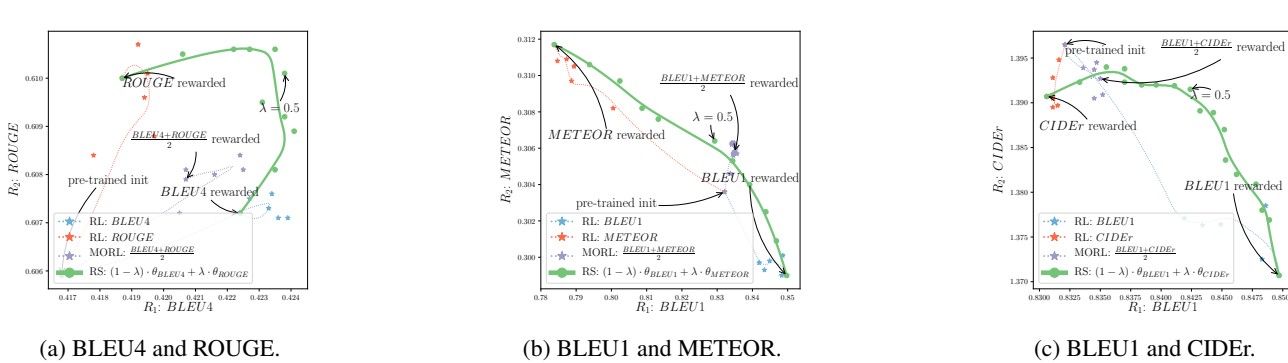

(a) BLEU4 and ROUGE.    (b) BLEU1 and METEOR.    (c) BLEU1 and CIDEr.

Figure 8: Additional results in captioning with more rewards, complementing Figure 3. Specifically, Figure 8(a) uses $R_1 = BLEU4$ and $R_2 = ROUGE$; then, with $R_1 = BLEU1$, Figure 8(b) uses $R_2 = METEOR$ and Figure 8(c) uses $R_2 = CIDEr$. In particular, the latter shows the failure when optimizing CIDEr; indeed, let's recall that the pre-trained initialization (Hu et al., 2022b) has actually already been trained by optimizing CIDEr (Rennie et al., 2017). Thus optimizing CIDEr a second time does not help, nor in CIDEr neither in other rewards. That's why in Figure 3(c) we consider the initialization as the network parametrization optimized for CIDEr.

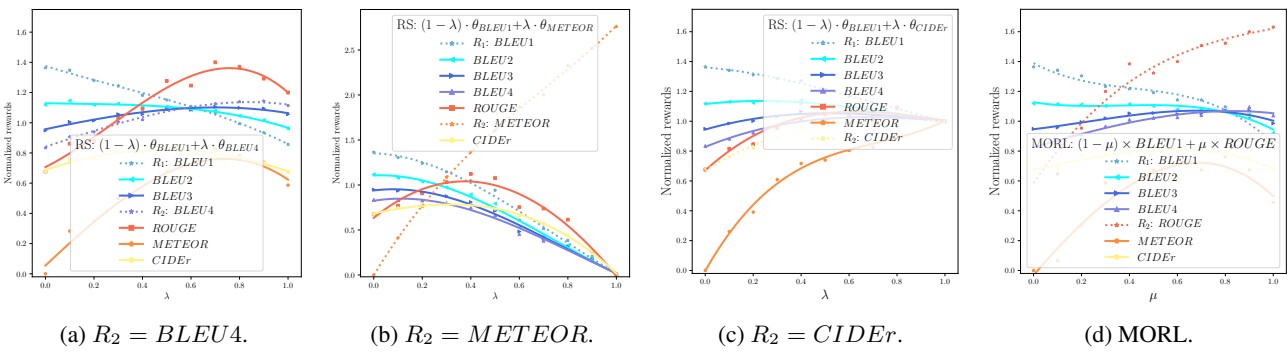

(a) $R_2 = BLEU4$.    (b) $R_2 = METEOR$.    (c) $R_2 = CIDEr$.    (d) MORL.

Figure 9: Additional results in captioning when measuring performances on all rewards and varying the interpolating coefficients, complementing Figure 4(b). In Figures 9(a) to 9(c), we extend the results for RS with $R_1 = BLEU1$ and for varying $R_2$; the optimal $\lambda$ depends on the similarity between the evaluation metric and $R_1$ and $R_2$. We also see in Figure 9(c) that all rewards are normalized to 1 for the CIDEr-initialization. In Figure 9(d), we perform the same analysis for MORL while varying the weighting $\mu$ over the proxy rewards $R_1 = BLEU1$ and $R_2 = ROUGE$; we recover similar curves than in Figure 4(b) for RS.

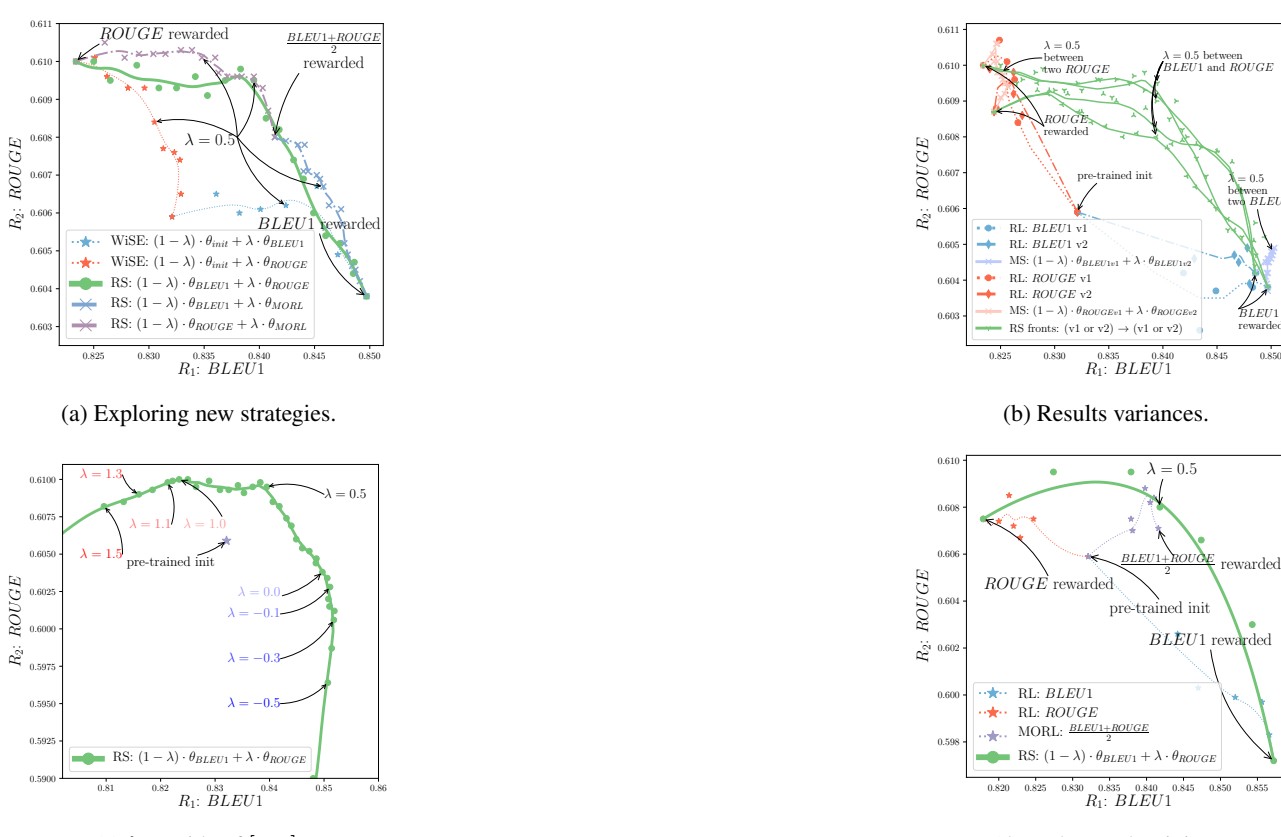

(a) Exploring new strategies.    (b) Results variances.

(c) $\lambda$ outside of $[0, 1]$.    (d) End-to-end training.

Figure 10: Additional results in captioning with $R_1 = BLEU1$ and $R_2 = ROUGE$. In Figure 10(a), we investigate interpolating the fine-tuned networks with the pre-trained initialization as in WiSE (Wortsman et al., 2022b); this only reveals a small portion of the front. In contrast, the interpolation with $\theta_{MORL}$ ($\mu = 0.5$) solution improves RS's front: this highlights some limitations in Hypothesis 2 and strict Pareto optimality of RS. Adding the MORL solutions as *intermediate* weights may help interpolate between two weights too distant. This suggests some practical complementarity between RS and MORL; given a training budget larger than the number of rewards, one may learn a few MORL for varying $0 \le \mu \le 1$, and then interpolate the obtained solutions. Figure 10(b) shows results' variance with two RL trainings for BLEU, and two for ROUGE, each time with a different seed defining the data ordering and augmentations. Though we observe some randomness, the Hypothesis 1 is consistently validated. Moreover, it presents the fronts described when we interpolate weights fine-tuned on a shared reward, as in model soups (MS) (Wortsman et al., 2022a; Ramé et al., 2022). This also only reveals a small portion of the spectrum of preferences, validating the need for diverse rewards to reveal the full Pareto front. In Figure 4(c), we report the front of the costly ensembling (Hansen & Salamon, 1990) of predictions (rather than of weights). Actually, ensembling performs better, but it cannot be fairly compared as its inference cost is doubled. Finally, Figure 10(d) shows the results when the networks are trained end-to-end, rather than keeping the backbone frozen. This validates the efficiency of rewarded soups in a new more general setting where all layers are trainable.

# E. Text-to-image: diffusion models with RLHFs

## E.1. Experimental details

Several works have studied the problem of aligning the output of diffusion models with human feedbacks (Lee et al., 2023; Wu et al., 2023b; Xu et al., 2023). Models are expected to understand specific visual control signals like colors, counts, and backgrounds more accurately after alignment. Notably, diffusion models can be fine-tuned to match human aesthetics preferences. As for any subjective metric, there is a variety of reward models that capture different aspects of aesthetic preference. These models are trained in a supervised setting to match human quality ratings collected on large image datasets (Xu et al., 2023), like the AVA dataset (Murray et al., 2012). As in the previous sections, RS allows to efficiently optimize multiple aesthetics reward models at test time, which allows adapting to the preferences of a single user.

We consider three metrics as rewards models: The cafe aesthetics model [1], trained on 3500 real-life and anime/manga images; An aesthetic score predictor based on CLIP features[2], trained on 250 000 images from the AVA dataset (Murray et al., 2012); we also experiment with a CLIP-based NSFW detector that estimates the probability of an image being "safe" by computing the cosine similarity with the embeddings of a set of "unsafe" words. The last two reward models are used to filter the LAION dataset (Schuhmann et al., 2021).

To fine-tune a diffusion model on a reward model $R$, we first generate 10000 images with the pre-trained diffusion model and compute the rewards for every generated image. Then, we fine-tune the diffusion model on the reward-weighted negative log-likelihood (Lee et al., 2023):

$$\mathcal{L} = \mathbb{E}_{(\mathbf{x}_0, Q) \in \mathcal{D}, \epsilon \sim \mathcal{N}(0,1), t \sim Uniform(0,T)} \quad r(\mathbf{x}_0) \|\epsilon_\theta(\mathbf{x}_t, t, Q) - \epsilon\|^2 \tag{19}$$

where $\epsilon_\theta$ is the noise estimation network, $T$ is the total number of training steps, $r(\mathbf{x}_0)$ is the reward of image $\mathbf{x}_0$ and $Q$ is the text associated to image $\mathbf{x}_0$.

On-policy Reinforcement Learning would normally require to perform loops of image generation and model fine-tuning (Dong et al., 2023), but we only perform a single optimization loop for simplicity.

**Implementation details.** We use a 2.2B parameters diffusion model trained on an internal dataset of 300M images, which reaches similar generation quality as Stable Diffusion (Rombach et al., 2022) in terms of CLIP alignment and FID scores. For efficient finetuning, we only fine-tune 10% of the diffusion model's weights (Xie et al., 2023) corresponding to the cross-attention layers and the bias/scaling parameters. For computational efficiency, we remove the 50% images with the worse scores, and rescale rewards linearly so that $\min_{\mathbf{x}_0 \in \mathcal{D}'} r(x_0) = 0$ and $\frac{1}{|\mathcal{D}'|} \sum_{\mathbf{x}_0 \in \mathcal{D}'} r(x_0) = 1$. All models are fine-tuned with Adam (Kingma & Ba, 2015) for 4000 steps with a batch size of 32 and learning rate 5e-6. Fine-tuned checkpoints and checkpoints interpolated with RS are evaluated on 1000 images.

Table 3: Image generation experiments: key implementation details.

| | Model |
|---|---|
| Architecture | GLIDE |
| Fine-tuning objective | Reward-weighted Diffusion Loss |
| Fine-tuning strategy | Fine-tuning cross-attention layers and bias/scale parameters |
| Optimizer | Adam (Kingma & Ba, 2015) |
| Dataset | Generated with COCO prompts |
| Rewards | *ava* (Murray et al., 2012) and *cafe* |
| Learning rate | 5e-6 |
| Batch size | 64 |
| Epochs | 25 |
| Hardware | Single GPU V100 32G |
| Compute budget | 500 GPUh |

---

[1]available at https://huggingface.co/cafeai/cafe_aesthetic
[2]available at https://github.com/christophschuhmann/improved-aesthetic-predictor/

## E.2. Additional results

We show in Figure 11 the spider map when computing MORL and RS on all three metrics: *ava*, *cafe* and the *nsfw* detector. In this case, MORL has higher scores than RS on the *ava* and *cafe* scores. We speculate that this is because the *nsfw* is very different from aesthetics preferences and that it can be inversely correlated with image quality: we have indeed noticed that lower quality images result in higher scores for the *nsfw* metric, being less often flagged as *unsafe*.

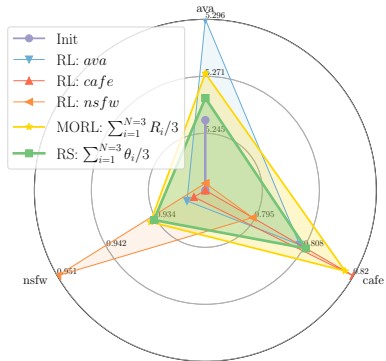

Figure 11: Image generation: spider map.

## F. Text-to-box: visual grounding

### F.1. Experimental details

We show the implementation details in Table 4. We use an internal unified model (Wang et al., 2022a; Lu et al., 2022) which will be released soon. The model is pre-trained solely on public benchmarks, to solve a variety of multimodal tasks such as VQA, visual grounding and image captioning. It is then fine-tuned on RefCOCO+ dataset for visual grounding. During the last fine-tuning phase, we complement the cross-entropy loss with an additional REINFORCE (Williams, 1992) term rewarding accuracy when the object is of the considered size. This means that the loss for $\theta_{Small}$ is $-\big(log(\hat{y}) + 5 \times 1_{\{\text{area}(\hat{y}) \text{ is small}\}} \times 1_{AUC(y,\hat{y})>0.5} \times log(y)\big)$ for an object with ground-truth box $\hat{y}$ and prediction $y$. The image is discretized into $1000 \times 1000$ bins before calculating the box areas. The task is illustrated in Figure 12.

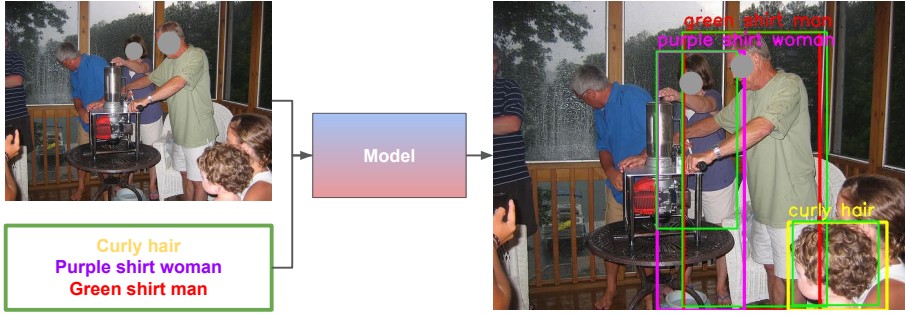

Figure 12: Illustration of the Visual Grounding task. The RS model results from the average of $N = 3$ weights specialized to detect respectively small, medium and large objects. The model takes the text (one at a time) as input and outputs the bounding box (i.e., colored predictions) in the image region described by the text (the ground truths are shown in green). We show an example of small, medium and large predictions. The texts and image input are from the validation set of RefCOCO+ (Yu et al., 2016).

Table 4: Visual grounding experiments: key implementation details.

| Model | |
|---|---|
| Architecture | Unified Model (ResNet-101+BART (Lewis et al., 2020)) |
| Visual encoder | ResNet-101 |
| Pretraining | Cross-Entropy on Public datasets (VQA, VG, Captioning) |
| Finetuning | Cross-Entropy on RefCOCO+ (Yu et al., 2016) |
| **RL procedure** | |
| Fine-tuning strategy | end-to-end |
| Dataset | RefCOCO+ (Yu et al., 2016) |
| RL algorithm | Cross-entropy + $5\times$ REINFORCE |
| Reward Small | IoU>0.5 for object with area $< 30000$ |
| Reward Medium | IoU>0.5 for object with $30000 \leq$ area $< 100000$ |
| Reward Large | IoU>0.5 for object with $100000 \leq$ area |
| Optimizer | Adam |
| Learning rate | 3e-5 |
| Batch size | 256 |
| Epochs | 10 |
| Hardware | 8 GPU 60GB |
| Compute budget | 800 GPUh |

## F.2. Additional results

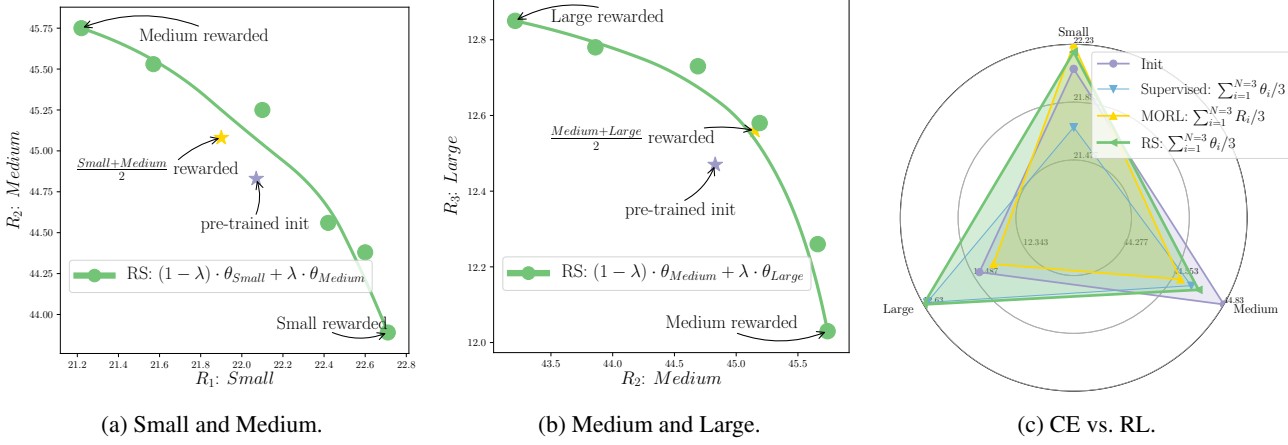

(a) Small and Medium.  (b) Medium and Large.  (c) CE vs. RL.

Figure 13: Results in visual grounding on RefCOCO+ (Yu et al., 2016). We use REINFORCE (Williams, 1992) to improve directly the non-differentiable accuracy, i.e., predict boxes with IoU$> 0.5$ w.r.t. the ground-truth. Trainings are specialized on either small, medium or large objects. These experiments complement Figures 5(b) and 5(c). Finally, Figure 13(c) compares between cross-entropy (CE) supervised fine-tuning (with Cross-entropy CE) and REINFORCE RL fine-tuning, using RS and MORL.

# G. Locomotion with diverse engineered rewards

## G.1. Experimental details

**Setup and task.** This experiment consists in fine-tuning a policy that has already learned how to make an humanoid run on Brax physics engine (Freeman et al., 2021).

**Pre-training.** We used the Brax implementation of PPO (Schulman et al., 2017) algorithm to pre-train the base policy used fine-tuning (see Table 5). The goal task used for pre-training is to make a Humanoid run with the default dense reward implemented in Brax: $R = velocity - 0.5 \cdot a_t^T a_t$. This phase is also used to collected statistics about observations and normalize them before inputting to the model, which helps training a lot.

**Fine-tuning.** The pre-trained policy is saved while the value function is discarded. We use the normalization procedure inherited from the pre-training but freeze it. We keep the same environment. Two reward functions are designed: a *risky* one for $R_1(t) = velocity$ and a *cautious* one where $R_2(t) = velocity - \cdot a_t^T a_t$. We make a grid-search on a few hyperparameters over 3 seeds (see the values between brackets in Table 5).

Table 5: Locomotion experiments: key implementation details.

| **PPO Pre-training** | |
| --- | --- |
| Interactions | 5e8 |
| Reward Scaling | 1.0 |
| Episode Length | 1000 |
| Normalize observations | True |
| Unroll Length | 10 |
| Discounting | 0.99 |
| Learning Rate | 5e-5 |
| Entropy Cost | 1e-3 |
| Number of environments in parallel | 4096 |
| Batch Size | 1024 |
| Hardware | 1GPU Tesla V100-SXM2-16GB |
| Runtime per experiment | 80min |
| **PPO Fine-tuning** | |
| Interactions | 1e8 |
| Reward Scaling | 1. |
| Normalize observations | True |
| Unroll Length | 10 |
| Discounting | {0.97, 0.99, 0.999} |
| Learning Rate | (1e-5, 3e-5, 1e-4) |
| Entropy Cost | 1e-3, 3e-3, 1e-2 |
| Number of environments in parallel | 4096 |
| Batch Size | 1024 |
| Hardware | 1GPU Tesla V100-SXM2-16GB |
| Runtime per experiment | 20min |
| **Model architecture** | |
| **Policy** | |
| Architecture | MLP |
| Nb of Layers | 6 |
| Hidden Size | 512 |
| **Value** | |
| Architecture | MLP |
| Nb of Layers | 5 |
| Hidden Size | 256 |

## G.2. Additional results

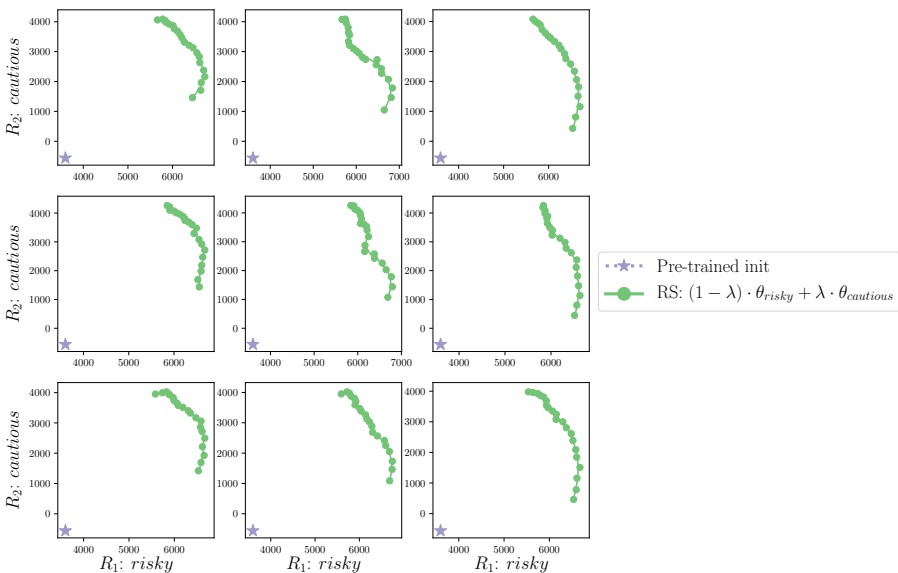

Figure 14: Some other runs for the locomotion task when varying the seed / hyperparameters.

