# OpenReview forum: "Rewarded soups: towards Pareto-optimal alignment by interpolating weights fine-tuned on diverse rewards"
_ICML.cc/2023/Workshop/ILHF — ILHF Workshop ICML 2023_

### Official Review · Reviewer_oyNx · 2023-06-11
**Review for Submission #9**

**Rating:** 6
**Confidence:** 5

**Review:**

The paper extends the single-reward setting in RLHF to multi-reward case, and studies the Pareto-optimal generalization across the space of preferences,  to handle the diversity of human preferences.

Compared with the standard multi-objective RL scalarization strategy, which interpolates reward during the fine-tuning of the policy, the proposed rewarded soup method directly trains N policies from N diverse reward functions, and interpolates the weights of the policies to get the final policy. This reduces the computational cost of the original MORL strategy. The authors have also conducted a large amount of experiments to validate the empirical performance of the MORL.

Overall, I think the paper is a good suit for ILHF program. My main concern is related to Working Hypothesis 1 (LMC). This suggests that the reward is a concave function of the weights for the policy network, which does not make sense to me since one can easily construct reward function that does not satisfy this property. It might be better for the authors to discuss clearly whether this really holds in all practical reward functions, and if it really holds, what's the hidden structure of reward and policy functions that lead to this property.

---

### Official Review · Reviewer_sGj3 · 2023-06-14
**Review from Reviewer sGj3**

**Rating:** 7
**Confidence:** 3

**Review:**

The work presented in this paper is well-motivated and the presentation is very clear. I was aware that the traditional strategy in multi-objective RL is to use different linear combination weights to transform the reward vector into a scalar reward or even sometimes combined with a more complicated genetic algorithm during the **training phase**.  However, this approach is not suitable for LLMs as it requires a large number of policies to form the Pareto frontier. Leveraging the connectivity property is a clever idea to reduce the burden on memory and computation resources. The experiments are also solid in supporting the main idea of this paper, with impressive results (even though MORL achieves comparable result with RS, RS is superior in terms of computation).

But I have to admit that if the linear mode connectivity condition is natural in the scenario of LLMs. Indeed, while there is rich literature on this phenomenon,  a thorough study of LLMs is still required. This is because, for LLMs, the behavior observed in a relatively small model (e..g 7B) may not be transferred to a large one. I think the authors have presented a preliminary empirical verification of it with LLaMA-7B and hope to see more empirical studies in the future.

Some other comments are as follows.

- I think the authors may want to present some sample outputs to demonstrate the behavior of the obtained model;
- Another suggestion is that you may change the output length on page 23 as I personally observe that the behavior of the models aligned by TRL-PPO (LLaMA-7B particularly) can be quite different with different output length, although they both achieve a high reward;

Another minor suggestion: the double subscripts ``i'' in Working Hypothesis 1 and 2 is a little bit confusing and you may want to modify them.

---

### Decision · Program_Chairs · 2023-06-20

Accept